# Radiative impact of improved global parameterisations of oceanic dry deposition of ozone and lightning-generated NO$_x$

Ashok K. Luhar[1], Ian E. Galbally[1], Matthew T. Woodhouse[1]

[1]CSIRO Oceans and Atmosphere, Aspendale, 3195, Australia

*Correspondence to*: Ashok K. Luhar (ashok.luhar@csiro.au)

**Abstract.** We investigate the radiative impact of recent process-based improvements to oceanic ozone (O$_3$) dry deposition parameterisation and empirical improvements to lightning-generated oxides of nitrogen (LNO$_x$) parameterisation by
conducting a 5-year simulation of the Australian Community Climate and Earth System Simulator – United Kingdom Chemistry and Aerosol (ACCESS-UKCA) global chemistry-climate model, with radiative effects of O$_3$, methane (CH$_4$) and aerosol included. Compared to the base parameterisations, the global consequences of the two improved parameterisations on atmospheric composition are dominated by the LNO$_x$ change (which increases the LNO$_x$ production from 4.8 to 6.9 Tg N yr$^{-1}$) and include (a) an increase in the O$_3$ column of 3.75 DU and this O$_3$ change is centred on the tropical upper troposphere where
O$_3$ is most effective as a radiative forcer; (b) a decrease of 0.64 years in the atmospheric lifetime of CH$_4$ due to an increase in hydroxyl radical, which corresponds to a decrease of 0.31 years in the CH$_4$ lifetime per Tg N yr$^{-1}$ change in LNO$_x$; (c) an increase of 6.7% in the column integrated condensation nuclei concentration; and (d) a slight increase in high-level cloud cover. The two combined parameterisation changes cause an increase of 86.3 mW m$^{-2}$ in the globally-averaged all-sky net downward top-of-atmosphere (TOA) radiative flux (which is akin to instantaneous radiative forcing), and only 5% of which
is due to the dry deposition parameterisation change. Other global radiative changes from the use of the two parameterisations together include an increase in the downward longwave radiation and a decrease in the downward shortwave radiation at the Earth's surface. The indirect effect of LNO$_x$ on aerosol and cloud cover can at least partly explain the differences in the downward shortwave flux at the surface. It is demonstrated that although the total global LNO$_x$ production may be the same, how LNO$_x$ is distributed spatially makes a difference to radiative transfer. We estimate that for a reported uncertainty range of
5 ± 3 Tg N yr$^{-1}$ in global estimates of LNO$_x$, the uncertainty in the net downward TOA radiation is ± 119 mW m$^{-2}$. The corresponding uncertainly in the atmospheric methane lifetime is ± 0.92 years. Thus, the value of LNO$_x$ used within a model will influence the effective radiative forcing and global warming potential of anthropogenic CH$_4$, and influence the results of climate scenario modelling.

# 1 Introduction

Apart from water vapour, carbon dioxide ($CO_2$), methane ($CH_4$), ozone ($O_3$), nitrous oxide ($N_2O$), and fluorinated gases are the principal greenhouse gases in the atmosphere. Being radiatively active, these gases play an important role in the Earth's energy budget and hence climate system. Together with aerosol, their concentrations govern the impedance to transfer and loss from the atmosphere of radiative energy. Radiative forcing is a change in the top-of-atmosphere (TOA) energy budget as a result of an imposed anthropogenic or natural perturbation (for example, changes in aerosol or greenhouse gas concentrations, in downwelling solar radiation, or in land use). The climate system responds to this change by cooling or warming. In the Sixth Assessment Report (AR6) of the United Nations Intergovernmental Panel on Climate Change (IPCC), an effective radiative forcing (ERF) is termed as a change in net downward radiative flux at the TOA following a perturbation, including effects of any adjustments in both tropospheric and stratospheric temperatures, water vapour, clouds, and some surface properties, for example surface albedo from vegetation changes, but excluding any changes due to the global surface air temperature change (Forster et al., 2021). According to the IPCC AR6, the total abundance-based anthropogenic ERF due to increases in long-lived well-mixed greenhouse gases over the years 1750–2019 is $3.32 \pm 0.29$ W m$^{-2}$, of which $2.16 \pm 0.26$ W m$^{-2}$ is due to $CO_2$, $0.54 \pm 0.11$ W m$^{-2}$ to $CH_4$, $0.21 \pm 0.03$ W m$^{-2}$ to $N_2O$ and $0.41 \pm 0.08$ W m$^{-2}$ to total halogens (Forster et al., 2021). Ozone is a short-lived climate forcer (SLCF), with a globally-averaged lifetime of about 22 days in the troposphere (Young et al., 2013), and its anthropogenic ERF for the same years is estimated to be $0.47 \pm 0.23$ W m$^{-2}$, almost all (95%) of which is due to tropospheric $O_3$ changes. Thus, tropospheric $O_3$ provides the third largest anthropogenic ERF, and overall $O_3$ represents about 16% of the net anthropogenic ERF of $2.72 \pm 0.76$ W m$^{-2}$, the latter includes an aerosol ERF of $-1.1 \pm 0.6$ W m$^{-2}$ (Forster et al., 2021).

Ozone interacts with down-welling and up-welling solar (or shortwave) and terrestrial (or longwave) radiation. Any changes in the atmospheric distribution of $O_3$ contribute to changes in its radiative impact. Compared to the long-lived and well-mixed greenhouse gases, $O_3$ exhibits a highly spatially inhomogeneous distribution in the troposphere because of its short chemical lifetime compared to transport timescales, and therefore, it has strong radiative effects on regional scales.

Ozone is not emitted directly into the atmosphere but is formed in both the stratosphere and troposphere by photochemical reactions involving natural and anthropogenic precursor species. Ozone is an oxidant as well as a precursor to the formation of hydroxyl (OH) and hydroperoxyl radicals which play a critical role in the tropospheric chemical cycles of many trace gases, e.g., $CH_4$ and carbon monoxide (CO), and the production of aerosol. The tropospheric budget of $O_3$ is governed by its production through the photochemical oxidation of $CH_4$, CO and non-methane volatile organic compounds (NMVOC) in the presence of oxides of nitrogen ($NO_x$); removal by several chemical reactions; removal by dry deposition at surface of the Earth; and the downward transport of $O_3$ from the stratosphere.

This paper extends recent work on improvements to the oceanic $O_3$ dry deposition parameterisation (Luhar et al., 2018) and lightning-generated $NO_x$ (referred to as $LNO_x$) parameterisation (Luhar et al., 2021) to investigate the impact on radiative

transfer of these improvements via the use of the global Australian Community Climate and Earth System Simulator – United Kingdom Chemistry and Aerosol (ACCESS-UKCA) chemistry-climate model. One of the primary purposes of improving these physico-chemical processes is to improve the overall performance of chemistry-climate and earth system models.

Dry deposition is a significant sink of $O_3$ (Galbally and Roy, 1980; Luhar et al., 2017; Clifton et al., 2020), affecting $O_3$ mixing ratio, and its long-range transport and lifetime. The improved dry deposition parameterisation by Luhar et al. (2018) is a mechanistic air–sea exchange scheme that accounts for the concurrent waterside processes of molecular diffusion, $O_3$ solubility, first-order chemical reaction of $O_3$ with dissolved iodide, and turbulent transfer. It is a significant improvement over the assumption in most chemical transport models that the controlling term of surface resistance in the scheme for dry deposition velocity of $O_3$ at the ocean surface is constant based on Wesely (1989) (see section 2.2), and results in a smaller averaged $O_3$ dry deposition velocity to the ocean (by a factor of 2–3), in better agreement with observations, and an increase in the tropospheric $O_3$ burden by 1.5% and a decrease in the methane lifetime by 0.8%.

Although $LNO_x$ accounts for only about 10% of the global $NO_x$ source, it has a disproportionately large contribution to the tropospheric burdens of $O_3$ and OH (Dahlmann et al., 2011; Murray, 2016). It's impact on $O_3$ concentration, the $CH_4$ lifetime (against loss by tropospheric OH) and aerosol in turn influences atmospheric radiative transfer. Schumann and Huntrieser (2007) report a large uncertainty range of $5 \pm 3$ Tg nitrogen (N) $yr^{-1}$ in the global amount of $LNO_x$ generated. Other estimates of global $LNO_x$ emissions include $6 \pm 2$ Tg N $yr^{-1}$ (Martin et al., 2007) and ~ 9 Tg N $yr^{-1}$ (Nault et al., 2017).

[As a side note, while we estimate the globally-averaged direct energy dissipated from lightning flashes to be only ~ 0.2 mW $m^{-2}$ (see the Supplement S1 for details), the radiative energy retained in the atmosphere due to the net impact of $LNO_x$ on $O_3$ production and $CH_4$ loss is ~ 40 mW $m^{-2}$ per Tg N $yr^{-1}$ produced due to lightning (see section 3.5), which implies a radiative impact of ~ 80 – 320 mW $m^{-2}$ corresponding to the above $LNO_x$ range of $5 \pm 3$ Tg N $yr^{-1}$. The atmospheric radiative change resulting from lightning is thus roughly three orders of magnitude larger than the direct energy release associated with the lightning flashes, a remarkable atmospheric amplifier.]

In most global chemistry models, lightning flash rates used to estimate $LNO_x$ are expressed in terms of convective cloud-top height via Price and Rind's (1992) (PR92) empirical parameterisations for land and ocean. Luhar et al. (2021) tested the PR92 flash-rate parameterisations within ACCESS-UKCA using satellite lightning data and found that while the PR92 parameterisation for land performs well, the oceanic parameterisation underestimates the observed global mean flash frequency by a factor of approximately 30, leading to $LNO_x$ being underestimated proportionally over the ocean. Luhar et al. (2021) improved upon the PR92 flash-rate parameterisations (see section 2.3). They showed that the improved parameterisation for land performs very similar to the corresponding PR92 one in simulating the continental spatial distribution of the global lightning flash rate. The improved oceanic parameterisation simulates the oceanic and total flash-rate observations much more accurately. Luhar et al. (2021) used the improved flash-rate parameterisations in ACCESS-UKCA and found that they resulted in a considerable impact on the modelled tropospheric composition compared to the default PR92 parameterisations, including

an increase in the global $LNO_x$ increased from 4.8 to 6.6 Tg N yr$^{-1}$; an increase in the ozone $O_3$ burden by 8.5%; a 13% increase in the volume-weighted global OH; and a decrease in the methane lifetime by 6.7%. The improved flash-rate parameterisations also led to improved simulation of tropospheric $NO_x$ and ozone in the Southern Hemisphere and over the ocean compared to observations. Luhar et al. (2021) did not examine any changes in aerosol due to the changes in $LNO_x$ (this is done in the present work).

We conduct a number of ACCESS-UKCA model simulations to quantify the effects of the above two parameterisation changes on the net downward shortwave (SW) and longwave (LW) radiative fluxes at TOA, and the downward surface SW and LW radiative fluxes, and these are reported here. Part of these changes arises due to the changes in atmospheric lifetime of $CH_4$, and that is included in the discussion. The calculated changes in the radiative fluxes are also put in the context of the IPCC anthropogenic ERF estimates.

## 2 The ACCESS-UKCA global chemistry-climate model and model setup

We use the United Kingdom Chemistry and Aerosol (UKCA) global atmospheric composition model (Abraham et al., 2012; https://www.ukca.ac.uk) coupled to ACCESS (Australian Community Climate and Earth System Simulator) (Bi et al., 2013; Woodhouse et al., 2015). In the simulations carried out here, ACCESS is essentially the same as the U.K. Met Office's Unified Model (UM) (vn8.4) as the ACCESS specific land-surface and ocean components are not invoked. The UM's original land-surface scheme (viz. JULES) is used and the model is run in atmosphere-only mode with prescribed monthly-mean sea surface temperature (SST) and sea ice fields. The atmosphere component of the UM vn8.4 is the Global Atmosphere (GA 4.0) (Walters et al., 2014). The UKCA configuration used here is the so-called StratTrop (or Chemistry of the Stratosphere and Troposphere (CheST)) (Archibald et al., 2020), which also includes the GLObal Model of Aerosol Processes (GLOMAP)-mode aerosol scheme (Mann et al., 2010). Dust is treated outside of GLOMAP-mode as per the scheme described by Woodward (2001).

The tropospheric chemistry scheme includes the chemical cycles of $O_x$, $HO_x$ and $NO_x$, and the oxidation of CO, $CH_4$, and other volatile organic carbon (VOC) species (for example, ethane, propane, and isoprene). The Fast-JX photolysis scheme reported by Neu et al. (2007) and Telford et al. (2013) is used. Ozone is coupled interactively between chemistry and radiation. The aerosol section includes sulphur chemistry. The total number of chemical reactions, including those in aerosol chemistry, is 306 across 86 species.

The horizontal resolution of the atmospheric model is 1.875° longitude × 1.25° latitude, with 85 staggered terrain-following hybrid-height levels extending from the surface to 85 km in altitude (the so-called N96L85 configuration). The vertical resolution becomes coarser with height, with the lowest 65 levels (altitudes up to ~ 30 km) located within the troposphere and lower stratosphere. The model's dynamical timestep is 20 min, and the UKCA chemical solver is called every 60 min.

A global monthly-varying emissions database for reactive gases and aerosol is used, which includes both anthropogenic, biomass burning and natural components (Woodhouse et al., 2015; Desservettaz et al., 2022). Pre-2000 anthropogenic

emissions are prescribed from the Atmospheric Chemistry and Climate Model Intercomparison Project (ACCMIP) (Lamarque et al., 2010), and post-2000 from Representative Concentrations Pathway (RCP) 6.0 scenario (van Vuuren et al. 2011). Biomass burning emissions are from the GFED4s database (van der Werf et al., 2017). Concentrations of $CO_2$, $CH_4$, $N_2O$ and $O_3$ depleting substances are prescribed instead of emissions and are from the Coupled Model Intercomparison Project Phase 5 (CMIP5) and RCP6.0 recommendations. Terrestrial biogenic emissions are from the Model of Emissions of Gases and Aerosols from Nature – Monitoring Atmospheric Composition and Climate dataset (MEGAN-MACC; Sindelarova et al., 2014), excepting soil $NO_x$ which is taken from the "global emissions initiative" (GEIA) project (https://www.geiacenter.org; last access 11 August 2014). The "present and future surface emissions of atmospheric compounds" (POET) database (Olivier et al., 2003) is used for oceanic ethane, propane, and CO emissions. Details of required emissions of other species, and their original sources, including biogenic emissions, chemical precursors and primary aerosol are given by Woodhouse et al. (2015).

## 2.1 Radiation scheme

UKCA is coupled to ACCESS's radiation scheme to determine the impact of the UKCA aerosol and radiatively active trace gases (normally $O_3$, $CH_4$, $N_2O$ and $O_3$ depleting substances) for any specific model configuration. For radiatively active trace gases, the Edwards and Slingo (1996) scheme, with updates described by Walters et al. (2014), is used. For the major gases (i.e., the dominant absorbers) in the shortwave bands, absorption by water vapour ($H_2O$), $O_3$, $CO_2$ and oxygen ($O_2$) is included. The treatment of $O_3$ absorption is as described by Zhong et al. (2008). For the major gases in the longwave bands, absorption by $H_2O$, $O_3$, $CO_2$, $CH_4$, $N_2O$, and halocarbons is included. The treatment of $CO_2$ and $O_3$ absorption is as described by Zhong and Haigh (2000). Of the major gases considered, $H_2O$ and $O_3$ are prognostic, whilst other gases are prescribed using either fixed or time-varying mass mixing ratios and assumed to be well mixed. The method of equivalent extinction (Edwards, 1996) is used for the minor gases (i.e., the weak absorbers) in each band.

In the present UKCA configuration, aerosol (direct scattering and absorption), $O_3$, $CH_4$, $N_2O$ are coupled to the radiation code where aerosol and $O_3$ are passed from the modelled 3-D fields. Aerosol additionally influences the large-scale cloud and precipitation schemes through the cloud droplet number concentration, whereas convective rainfall and cloud formation are not directly coupled to the model aerosol scheme but can be indirectly influenced via changes in radiation which can in turn influence properties such as temperature and moisture (Abraham et al., 2012; Bellouin et al., 2013; Fiddes et al., 2018).

$LNO_x$ is also a precursor of nitrate aerosol in the upper troposphere, and this aerosol can influence atmospheric radiation (Tost, 2017). However, ACCESS-UKCA as used here does not include nitrate aerosol, which is also the case with most global chemistry-climate models. Of the ten CMIP6 Earth system models that conducted the AerChemMIP (Aerosol and Chemistry Model Intercomparison Project) simulations, only three included nitrate aerosols (Thornhill et al., 2021b). Naik et al. (2021) report that there is a relatively small negative contribution to ERF through formation of nitrate aerosols. Recently, a nitrate scheme has been incorporated in UKCA (Jones et al., 2021) and this should be tested in the future to examine the impact of nitrate aerosol from lightning on radiation. Although the model does not include a nitrate aerosol scheme, the $LNO_x$ changes

would impact aerosol through perturbations to background tropospheric oxidants, for example increases in aerosol abundances due to faster oxidation rates of sulfur to sulfate as $LNO_x$ is increased (Murray, 2016).

## 2.2 Ozone dry deposition scheme for the ocean

Dry deposition flux of $O_3$ to Earth's surface is modelled as the product of $O_3$ concentration in the air near the surface and a (downward) dry deposition velocity, $v_d$, which is calculated as (Wesely, 1989)

$$v_d = \frac{1}{r_a + r_b + r_c},$$ (1)

where $r_a$ is the aerodynamic resistance which is the resistance to transfer by turbulence in the atmospheric surface layer, $r_b$ is the atmospheric viscous (or quasi laminar) sublayer resistance which is the resistance to movement across a thin layer (0.1 – 1 mm) of air that is in direct contact with the surface, and $r_c$ is the surface resistance which is the resistance to uptake by the surface itself. Various parameterisations are used to calculate these resistances. $r_c$ is the dominant term in Eq. (1) for $O_3$ dry deposition to water surfaces, and it is routinely assumed that $r_c$ for water is constant at $\approx 2000$ s m$^{-1}$ following Wesely's (1989) widely used dry deposition parameterisation. Most global chemical transport models, e.g. CAM-chem (Lamarque et al., 2012), GEOS-Chem (Mao et al., 2013) and UKCA, have followed this approach thus far by default, with ACCESS-UKCA using $r_c = 2200$ s m$^{-1}$.

Recently, Luhar et al. (2017, 2018) concluded that the use of the above constant $r_c$ approach in ACCESS-UKCA overestimates $O_3$ deposition velocities to the ocean by as much as a factor of 2 to 4 compared to measurements, and does not simulate their observed spatial variability well. Luhar et al. (2018) developed a two-layer process-based parameterisation for $r_c$ that accounts for the concurrent waterside processes of molecular diffusion, $O_3$ solubility, first-order chemical reaction of $O_3$ with dissolved iodide, and turbulent transfer, and found that this parameterisation described the $O_3$ deposition velocities much better and reduced the global oceanic $O_3$ deposition to approximately one-third of the default value obtained using Wesely's (1989) $r_c$ approach. Using the new/improved parameterisation, Luhar et al. (2018) estimated an oceanic dry deposition of $98.4 \pm 30.0$ Tg $O_3$ yr$^{-1}$ and a global one of $722.8 \pm 87.3$ Tg $O_3$ yr$^{-1}$ (averaged over years 2003–2012), which can be compared with the respective values 340 Tg $O_3$ yr$^{-1}$ and $978 \pm 127$ Tg $O_3$ yr$^{-1}$ obtained by Hardacre et al. (2015) based on 15 global chemistry transport models (for year 2001) using Wesely's scheme, demonstrating the large reduction in the oceanic value using the new parameterisation. The new approach has recently been evaluated by other researchers in both global and regional models with various changes to input parameter values (Loades et al., 2020; Pound et al., 2020; Barten et al., 2021).

We use both the default and new oceanic dry deposition parameterisations (the latter corresponding to the Ranking 1 configuration in Table one of Luhar et al. (2018)).

## 2.3 Lightning-generated NO$_x$

NO$_x$, which is a mixture of nitrogen dioxide (NO$_2$) and nitric oxide (NO), acts as a precursor to O$_3$ and OH, which are the principal tropospheric oxidants. Lightning mainly happens in the tropics related to deep atmospheric convection and is the primary source of NO$_x$ in the middle to upper troposphere where lightning is mostly discharged. A tropospheric ozone radiative

kernel for all-sky conditions (i.e., clear, cloud overcast, and partially cloudy skies) derived by Rap et al. (2015) suggests that ozone changes in the tropical upper troposphere are up to 10 times more efficient in altering the Earth's radiative flux than other regions.

As stated earlier, Schumann and Huntrieser (2007) report a range of $5 \pm 3$ Tg nitrogen (N) yr$^{-1}$ produced by lightning globally. The range of global LNO$_x$ in 16 ACCMIP models in CMIP5 varied between 1.2 to 9.7 Tg N yr$^{-1}$ (Lamarque et al., 2013),

whereas in five earth system models in the Coupled Model Intercomparison Project Phase 6 (CMIP6) LNO$_x$ ranged between 3.2 to 7.6 Tg N yr$^{-1}$ (Griffiths et al., 2021; Naik et al., 2021) for the present-day (nominal year 2000) conditions.

The LNO$_x$ amount in most global models is calculated as

$$LNO_x = P_{NO} \times F,$$

(2)

where $P_{NO}$ is the quantity of NO generated per lightning flash, and $F$ is the flash rate. $F$ is calculated at every model time step within a model grid, and partitioned into cloud-to-ground (CG) and intracloud (IC) flash components. An emission factor of

the amount of NO generated per CG/IC flash is applied, and the calculated mass of NO is then distributed vertically in the grid column (Luhar et al., 2021).

Of all the past techniques used to determine lightning flash rate in global chemistry-climate models and chemical transport models, including ACCESS-UKCA, the PR92 parameterisations are the most commonly used ones. They (or very similar) have also been used in most CMIP5 and CMIP6 models.

The PR92 parameterisations for lightning flash rate (flashes per minute) over land ($F_L$) and ocean ($F_O$) are

$$F_L = 3.44 \times 10^{-5} \, H^{4.9},$$

(3)

$$F_O = 6.4 \times 10^{-4} \, H^{1.73},$$

(4)

where $H$ is the height of the convective cloud top (km), which is passed from the model's convection parameterisation scheme. The above parameterisations yield flash rates over the ocean that are smaller by approximately 2 to 3 orders of magnitude compared to those calculated for clouds over land.

The oceanic parameterisation Eq. (4) is known to greatly underestimate flash rates. Recently, Luhar et al. (2021) evaluated the

PR92 parameterisations for the year 2006 and found that while the land parameterisation Eq. (3) gave satisfactory predictions (an average value of 32.5 flashes s$^{-1}$ compared to 34.9 flashes s$^{-1}$ obtained from satellite observations), the oceanic

parameterisation Eq. (4) yielded a global mean value of 0.33 flashes s⁻¹ over the ocean, a much smaller value than the observed 9.16 flashes s⁻¹. They formulated the following improved flash-rate parameterisations using the scaling relationships between thunderstorm electrical generator power and storm geometry developed by Boccippio's (2002), together with available data:

$$F_L = 2.40 \times 10^{-5} H^{5.09}, \tag{5}$$

$$F_O = 2.0 \times 10^{-5} H^{4.38}. \tag{6}$$

Flash rates obtained using Eq. (6) are approximately two orders of magnitude greater than those obtained using Eq. (4). Eq. (5) performed very similar to Eq. (3), giving an average value of 35.9 flashes s⁻¹ compared to 34.9 flashes s⁻¹ obtained from satellite observations, and the new/improved marine parameterisation Eq. (6) gave a global mean marine flash rate of 8.84 flashes s⁻¹, which is very close to the observed value of 9.16 flashes s⁻¹.

With $P_{NO} = 330$ moles NO per flash, the use of Eqs. (5) and (6) in ACCESS-UKCA increased the mean total global LNO$_x$ by 38% from the base value of 4.8 Tg N yr⁻¹ (Luhar et al. (2021)), with a considerable impact on the tropospheric composition as stated in Section 1.

We investigate the radiative effects of this change in LNO$_x$.

## 2.4 Global model simulations

We conducted the following six ACCESS-UKCA simulations for the years 2004–2010. Considering the first two simulation years as model spin-up time, the output from the model for the five-year period 2006–2010 was used for the analysis reported below:

- Base run (Run A): Default model run, with $r_c = 2200$ s m⁻¹ in the oceanic O$_3$ deposition, and the PR92 lightning flash-rate parameterisation (LNO$_x$ = 4.8 Tg N yr⁻¹),

- Run B: New process-based oceanic O$_3$ deposition scheme and the PR92 lightning flash-rate parameterisation (LNO$_x$ = 4.8 Tg N yr⁻¹),

- Run C: New process-based oceanic O$_3$ deposition scheme and new lightning flash-rate parameterisation (LNO$_x$ = 6.9 Tg N yr⁻¹),

- Run D: New process-based oceanic O$_3$ deposition scheme, and the PR92 lightning flash-rate parameterisation but scaled uniformly by a factor of 1.44 to give the total global LNO$_x$ the same as Run C (LNO$_x$ = 6.9 Tg N yr⁻¹) (to check the impact of the difference in spatial distribution of the lightning flashes),

- Run E: Same as Run C, but without the CH$_4$ radiation feedback (to quantify its individual radiative impact) (LNO$_x$ = 6.9 Tg N yr⁻¹), and

- Run F: New process-based oceanic O$_3$ deposition scheme and LNO$_x$ = 0.

Apart from the above changes, the rest of the model setup is the same as described in Section 2 of Luhar et al. (2021). The simulations were nudged to the ECMWF's ERA-Interim reanalyses in the free troposphere involving horizontal wind components and potential temperature given on pressure levels at 6-hourly intervals (Dee et al., 2011; https://www.ecmwf.int/en/forecasts/datasets/reanalysis-datasets/era-interim). Each model run was initialised using a previously spun-up model output with nudging and the default lightning and dry deposition schemes. The use of nudging does not allow the model changes to adjust synoptic-scale meteorology; hence the results here represent instantaneous radiative responses in the climate system, unlike the ERF which is the sum of the instantaneous radiative forcing (IRF) and the contribution from such adjustments. Due to nudging, responses in the simulation may be dampened, but can be attributed directly to the model perturbations (Fiddes et al., 2018).

The model results were averaged over 5-years for the globe, tropics, extra-tropics, land, and sea. Differences between the base model run and the other runs were calculated and indexed as follows on the x-axis in relevant plots presented below.

- 1 (dep.)                          = Run B – Base
- 2 (dep. + $LNO_x$)                = Run C – Base
- 3 (dep. + scaled $LNO_x$)         = Run D – Base
- 4 (dep. + $LNO_x$ + no $CH_4$) = Run E – Base
- 5 (dep. + no $LNO_x$)             = Run F – Base

## 3 Results and discussion

Monthly-averaged model output for various radiative components and chemical species is used in the following.

### 3.1 Modelled ozone, methane lifetime and aerosol

With the new oceanic dry deposition scheme, the total global $O_3$ dry deposition is decreased by 12.3% (which is nearly the same as the 12.1% decrease obtained by Pound et al. (2020) using this scheme in GEOS-Chem), and the improved lightning parameterisations increase the global $LNO_x$ by 44% from 4.8 to 6.9 Tg N $yr^{-1}$. Table 1 summarises the global-averaged impact of the various parameterisation changes in ACCESS-UKCA. With the new dry deposition scheme (Run B), the global tropospheric $O_3$ burden increases by 1.5% over the base run. With both the new dry deposition and $LNO_x$ schemes (Run C) this increase in the tropospheric $O_3$ burden is by 11.7%. Similarly, the increase in the total $O_3$ column is by 0.14 DU for the new dry deposition scheme and 3.75 DU (13% of the tropospheric column or 1.2% of the total column) for the new dry deposition and $LNO_x$ schemes combined. The global distribution of the $O_3$ column difference between Run C and the base run in Figure 1a shows that the biggest differences as high as ≈ 8 DU occur in the tropics between 140°W –100°E.

Changes in $LNO_x$ and $O_3$ also affect the global mean lifetime of $CH_4$ due to loss by OH ($\tau_{CH_4}$) in the troposphere (Labrador et al., 2004). As Table 1 shows, there is a relatively small decrease of 0.07 years (0.9%) in $\tau_{CH_4}$ (corresponding to an increase of OH by 0.6%) when the new oceanic $O_3$ dry deposition scheme is used and this decrease is 0.64 years (8.4%) (corresponding

to an increase of OH by 15.6%) when the improved LNO$_x$ parameterisation is also used. In Table 1, the modelled methane lifetimes are lower than the ACCMIP multi-model mean 9.7 ± 1.5 years reported by Naik et al. (2013), which, as pointed out by Luhar et al. (2021), could be due to a higher tropospheric burden of non-lightning related NO$_x$ in ACCESS-UKCA and/or a more intense photolysis. However, because we are mainly focusing on differences with respect to the base run, the lower absolute values of $\tau_{CH_4}$ from ACCESS-UKCA are not considered to be as pertinent.

Table 1 also presents the modelled global-averaged column integrated condensation nuclei (CN, > 3 nm dry diameter, also denoted as $N_3$) or aerosol number concentration. The column CN concentration increases with LNO$_x$ and this increase for Run C is 6.7% over the base run. The global distribution of the CN column difference between Run C and the base run in Figure 1b shows that the biggest increases by as much as $5 \times 10^9$ cm$^{-2}$ occur in the tropics over the Atlantic Ocean. (Changes in cloud cover are reported in section 3.4.)

Results from the other model simulations are also presented in Table 1. The only difference between Run C and Run D is how the lightning flash rate is spatially distributed, and it is apparent that that make a significant difference in the results. As expected, the values from the simulations with and without the CH$_4$ radiation feedbacks (Runs C and E) are virtually the same.

**Table 1: Global-averaged values obtained from various model runs (for the period 2006–2010).**

| Model run | LNO$_x$ (Tg N yr$^{-1}$) | Tropospheric O$_3$ burden (Tg O$_3$) | Total/tropospheric O$_3$ column (DU) | OH ($10^6$ molec cm$^{-3}$) | CH$_4$ lifetime (yr) | CN column ($10^9$ cm$^{-2}$) |
|---|---|---|---|---|---|---|
| Run A (base) | 4.8 | 271.8 | 304.6/27.8 | 1.052 | 7.61 | 7.419 |
| Run B | 4.8 | 276.0 | 304.8/28.2 | 1.058 | 7.54 | 7.409 |
| Run C | 6.9 | 303.7 | 308.4/30.9 | 1.217 | 6.97 | 7.919 |
| Run D | 6.9 | 294.3 | 307.2/30.0 | 1.151 | 7.17 | 7.577 |
| Run E | 6.9 | 303.6 | 308.2/30.9 | 1.217 | 6.97 | 7.918 |
| Run F | 0 | 214.8 | 296.0/22.3 | 0.755 | 9.26 | 6.546 |

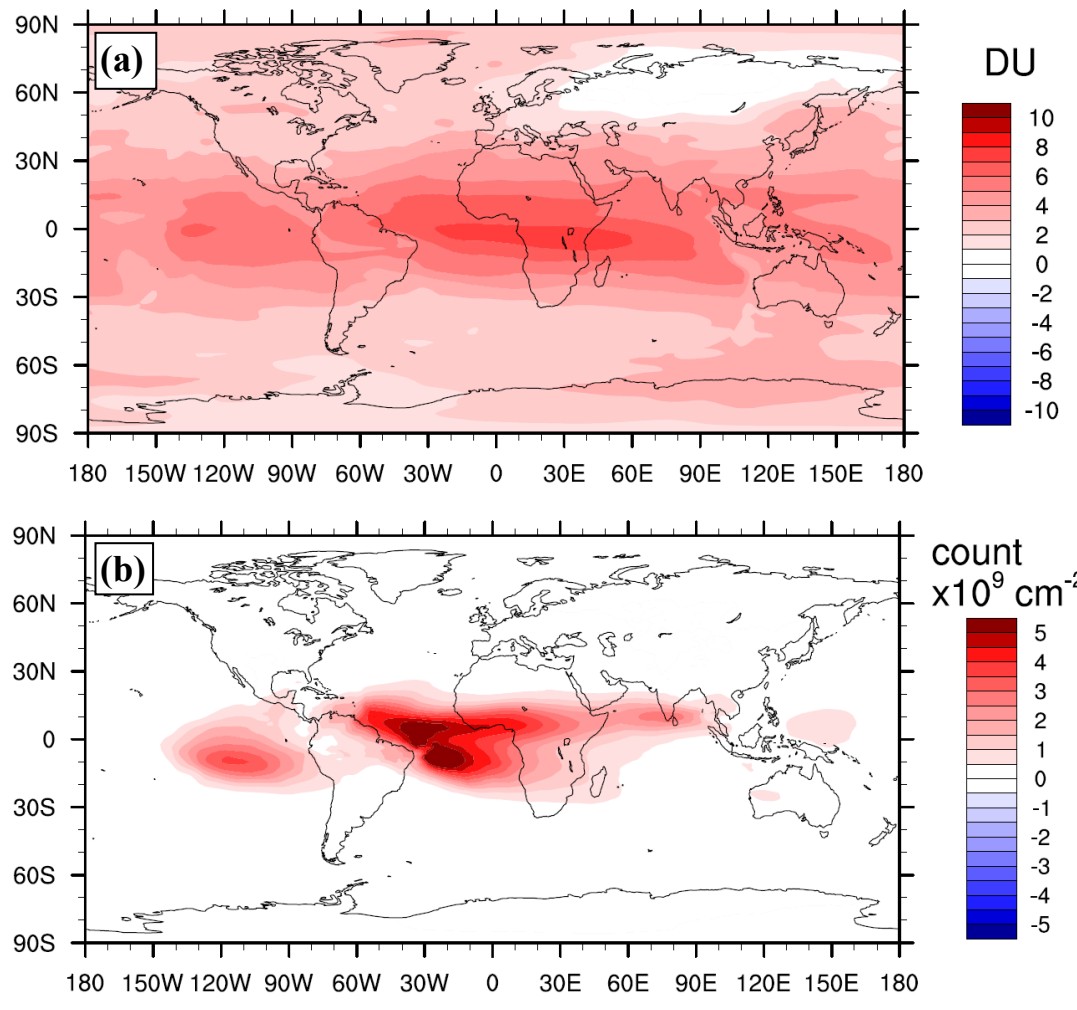

**Figure 1: (a) Modelled column ozone difference (DU), and (b) column condensation nuclei (CN) concentration difference ($10^9$ count cm$^{-2}$) between Run C (i.e., the new dry deposition scheme and the new LNO$_x$ scheme) and the base run (mean over 2006–2010).**

### 3.2 Model performance for all-sky radiative fluxes

Model output for radiative components for all-sky conditions includes the incoming (or downward) TOA solar radiation ($S\downarrow_{TOA}$) (which is the same in all model runs), outgoing (or upward) TOA longwave radiation ($L\uparrow_{TOA}$), outgoing TOA shortwave radiation ($S\uparrow_{TOA}$), incoming longwave radiation at the surface ($L\downarrow_S$), incoming shortwave radiation at the surface

10 ($S\downarrow_S$), outgoing longwave radiation at the surface ($L\uparrow_S$), and outgoing shortwave radiation at the surface ($S\uparrow_S$). The total radiation is the sum of the longwave and shortwave components.

The net downward TOA radiative flux is

$$R^N_{TOA} = S\downarrow_{TOA} - (L\uparrow_{TOA} + S\uparrow_{TOA}), \qquad (7)$$

where the superscript N signifies net (note that $S\downarrow_{TOA}$ is the same in all runs and $L\downarrow_{TOA} = 0$). Additional definitions are: the net downward TOA shortwave radiative flux $S^N_{TOA} = S\downarrow_{TOA} - S\uparrow_{TOA}$ and the net downward TOA longwave radiative flux $L^N_{TOA} = -L\uparrow_{TOA}$. The net downward longwave radiation at the surface (*rls*) is equal to $(L\downarrow_S - L\uparrow_S)$ and net downward shortwave radiation at the surface (*rss*) is equal to $(S\downarrow_S - S\uparrow_S)$. (Here the variable names in italics are based on standard CMIP convention used in model codes.) Table 2 gives a list of radiative flux symbols used. Model output for clear-sky $L\uparrow_{TOA}$ (*rlutcs*), $S\uparrow_{TOA}$ (*rsutcs*), $L\downarrow_S$ (*rldscs*), $S\downarrow_S$ (*rsdscs*) and $S\uparrow_S$ (*rsuscs*) was also available. The clear-sky fluxes were calculated using "method II" whereby at every grid-point the radiative flux is calculated using exactly the same physical inputs (gaseous mixing ratios, surface albedos etc.) as for the all-sky calculation, except that the radiative effects of clouds are ignored. Unless stated otherwise the reported radiative fluxes are for all-sky conditions.

**Table 2: List of radiative flux symbols for all-sky conditions.**

| Symbol | Definition |
|---|---|
| $L\uparrow_{TOA}$ | Outgoing (or upward) top-of-atmosphere longwave radiative flux (*rlut*) |
| $L\downarrow_{TOA}$ | Incoming (or downward) top-of-atmosphere longwave radiative flux (= 0) |
| $S\downarrow_{TOA}$ | Incoming top-of-atmosphere shortwave (or solar) radiative flux (*rsdt*) |
| $S\uparrow_{TOA}$ | Outgoing top-of-atmosphere shortwave radiative flux (*rsut*) |
| $L\downarrow_S$ | Incoming longwave radiative flux at the surface (*rlds*) |
| $S\downarrow_S$ | Incoming shortwave radiative flux at the surface (*rsds*) |
| $L\uparrow_S$ | Outgoing longwave radiative flux at the surface |
| $S\uparrow_S$ | Outgoing shortwave radiative flux at the surface |
| $R^N_{TOA}$ | Net downward top-of-atmosphere radiative flux (= $S^N_{TOA} + L^N_{TOA}$) |
| $S^N_{TOA}$ | Net downward top-of-atmosphere shortwave radiative flux |
| $L^N_{TOA}$ | Net downward top-of-atmosphere longwave radiative flux |

In Table 3, the area-weighted globally-averaged modelled radiative fluxes from the base ACCESS-UKCA run are in good agreement with the observed values computed from NASA's Clouds and the Earth's Radiant Energy System (CERES) EBAF (Energy Balanced and Filled) Ed4.1 dataset (https://ceres.larc.nasa.gov/data/) for the period 2006–2010 (Loeb et al., 2018; Kato et al., 2018), and with those from a 16-model ensemble from CMIP5 twentieth-century experiments (Stephens et al., 2012) (the range in terms of model minimum and maximum values is given in parenthesis).

**Table 3: Comparison of the modelled (base run) and observed radiative fluxes (W m⁻²) for all-sky conditions. The values are global averages for the period 2006–2010 and the values in parenthesis are minimum and maximum values.**

| Radiative flux | Modelled | Observed | CMIP5 |
|---|---|---|---|
| $S{\downarrow}_{TOA}$ | 341.44 | 339.93 | 343.0 (338.6–343.7) |
| $L{\uparrow}_{TOA}$ | 240.61 | 240.10 | 238.6 (232.4–243.5) |
| $S{\uparrow}_{TOA}$ | 102.24 | 99.19 | 102.2 (96.4–106.5) |
| $L{\downarrow}_{S}$ | 341.55 | 344.57 | 339.7 (326.4–347.0) |
| $S{\downarrow}_{S}$ | 191.49 | 186.48 | 190.3 (181.9–196.2) |
| $L{\uparrow}_{S}$ | 400.07 | 398.18 | 397.5 (391.9–398.1) |
| $S{\uparrow}_{S}$ | 24.83 | 23.21 | 24.9 (21.1–30.3) |

A comparison of the zonal-averaged modelled $L{\uparrow}_{TOA} + S{\uparrow}_{TOA}$, $L{\downarrow}_{S}$ and $S{\downarrow}_{S}$ with the corresponding CERES-EBAF data is made in Figure 4 and discussed in section 3.3.

Fiddes et al. (2018) obtained similar evaluation results for a very similar setup of ACCESS-UKCA vn8.4 for radiation components averaged over the period of 2000–2009.

**3.3 Radiative effects of the parameterisation changes**

Figure 2 presents bar charts of the modelled mean (2006–2010) difference ($\Delta$) in the all-sky, area-weighted net downward TOA total radiation ($\Delta R^{N}_{TOA}$) and its longwave ($\Delta L^{N}_{TOA}$) and shortwave ($\Delta S^{N}_{TOA}$) components between the various runs and the base run for the globe, tropics (here $\leq |30°|$), extra-tropics ($> |30°|$), land, and ocean (which includes all water bodies). In The absolute values of the total outgoing TOA radiative flux ($L{\uparrow}_{TOA} + S{\uparrow}_{TOA}$), and $L{\uparrow}_{TOA}$ and $S{\uparrow}_{TOA}$ for the base run are also plotted as a reference (corresponding to the right y-axes). Table 4 presents values of $\Delta R^{N}_{TOA}$, $\Delta L^{N}_{TOA}$, $\Delta S^{N}_{TOA}$, $\Delta L{\downarrow}_{S}$ and $\Delta S{\downarrow}_{S}$ for the globe, tropics, and extra-tropics for all-sky conditions (Table S1 the Supplement gives additional modelled flux differences). $\Delta R^{N}_{TOA}$ is akin to radiative forcing; a positive $\Delta R^{N}_{TOA}$ means more radiation is retained in the atmosphere due to perturbation to the base model. It is apparent that with the new oceanic deposition scheme, $R^{N}_{TOA}$ is increased, but by only a relatively very small amount of 4.4 mW m⁻². This small increase is due to an increase in the shortwave component which dominates over a decrease in the longwave component. With the new lightning flash-rate parameterisation also included this change in $R^{N}_{TOA}$ is much greater at 86.3 mW m⁻² as $LNO_x$ increases to 6.9 from the base value 4.8 Tg N yr⁻¹ with enhanced tropospheric $O_3$ production. This increase in $R^{N}_{TOA}$ is due to an increase in both longwave and shortwave components, but the former dominates. Increased $LNO_x$ causes enhanced OH concentrations that reduce the tropospheric $CH_4$ lifetime and would

increase $L\uparrow_{TOA}$. If we turn off the radiative feedbacks of $CH_4$ in the model, the increase in $R^N_{TOA}$ is 107.0 mW m$^{-2}$. In other words, the $CH_4$ feedback negates the positive radiative effect of $O_3$ feedback by 20.7 mW m$^{-2}$. (Note that dry deposition is a surface process and, therefore, any changes to it would influence $O_3$ in surface air to much greater extent than that at higher altitudes, so the radiative effects of these changes may be very small.)

5 When the default PR92 lightning flash-rate scheme is used with a uniform global scaling (by a factor of 1.44) so as to give the total global $LNO_x$ the same as that obtained by Run C with the new lightning flash-rate scheme (i.e., 6.9 Tg N yr$^{-1}$), the increase in $R^N_{TOA}$ is 70.9 mW m$^{-2}$. As stated earlier, while the PR92 scheme for land performs very similar to the new scheme in simulating the global spatial distribution of lightning flash rate over land, the oceanic PR92 scheme underestimates the global mean flash-rate distribution considerably over the ocean. Therefore, this uniform scaling of the PR92-derived global flash-rate

10 distribution would cause an over-adjustment of the flash rate (and hence $LNO_x$) over land to compensate for the underestimation by the oceanic parameterisation. Therefore, although the total global $LNO_x$ is the same in both Runs C and D, there is a mismatch in its spatial distribution with Run D having larger $LNO_x$ over land and continue to have lower $LNO_x$ over the ocean than Run C. Thus, the new lightning flash-rate scheme leading to a larger increase in $R^N_{TOA}$ than that obtained by the scaled PR92 scheme implies that how $LNO_x$ is spatially distributed makes a difference in the radiation impact. This difference

15 could possibly be because adding $LNO_x$ to the lower $NO_x$ levels in the marine upper troposphere causes greater ozone production than adding it to the $NO_x$ richer continental upper troposphere, and also because of differences in the photochemical reaction rates as a result of temperature differences over land and sea.

Turning off $LNO_x$ completely in the model causes an extra 190.8 mW m$^{-2}$ to leave the atmosphere compared to the base run. This relates to the lower amount of $O_3$ due to the absence of $LNO_x$ in the atmosphere. This increase in outgoing radiation

20 related to the reduced upper tropospheric $O_3$ dominates over the reverse radiative impact of no $LNO_x$ causing lower OH and hence a longer $CH_4$ lifetime.

In Figure 2a, the all-sky radiation changes relative to the base run are larger in magnitude in the tropics than elsewhere. With the new oceanic deposition scheme, the net downward TOA radiation is increased by 13.7 mW m$^{-2}$ in the tropics but is reduced by 5.5 mW m$^{-2}$ elsewhere. The contrast in radiation changes over land and ocean is not as stark as that over the tropical and

25 extra-tropical regions, except for the no-$LNO_x$ case.

Figure 2b and Figure 2c are the same as Figure 2a except that they are for the differences in the net downward TOA radiation components $L^N_{TOA}$ and $S^N_{TOA}$, respectively (Table 4 gives the values). These plots suggest that when the new $LNO_x$ scheme is used, the changes in the total net downward TOA radiation are dominated by the changes in the longwave component ($L^N_{TOA}$).

Figure 3a is a bar chart of the modelled mean difference of the area-weighted downward surface longwave radiative flux

30 ($\Delta L\downarrow_S$) between the various runs and the base run. The absolute values of the surface longwave radiative flux for the base run are also plotted as a reference. Figure 3b is the corresponding plot for the downward surface shortwave radiative flux difference ($\Delta S\downarrow_S$). The $LNO_x$ increases in the model, compared to the base run, lead to an increase in $L\downarrow_S$, and a decrease in $S\downarrow_S$.

Table 4 also presents the mean global radiative flux differences for clear-sky conditions (Table S2 in the Supplement gives additional clear-sky modelled flux differences). For the case when the new deposition and lightning flash-rate parameterisations are used, the clear-sky $\Delta R^N_{TOA}$ is greater at 110.8 mW m$^{-2}$ compared to the all-sky value of 86.3 mW m$^{-2}$, and in both cases this change is dominated by the longwave component. For the case when only the new ozone dry deposition scheme is used, the clear-sky $\Delta R^N_{TOA}$ is again relatively very small at -6 mW m$^{-2}$ (driven by a decrease in the longwave component) but is in opposite direction compared to the corresponding all-sky case. The clear-sky mean global increases in the surface L$\downarrow_S$ are larger compared to the all-sky increases. Thus, while the all-sky and clear-sky radiations show qualitatively similar changes in response to changes in LNO$_x$, the differences in the magnitude of these changes imply an impact of LNO$_x$ on clouds and an analysis on this is given in section 3.4.

**Table 4: Changes in the modelled net downward total TOA radiative flux ($\Delta R_{TOA}^{N}$), net downward TOA longwave radiative flux ($\Delta L_{TOA}^{N}$), net downward TOA shortwave radiative flux ($\Delta S_{TOA}^{N}$), and incoming longwave ($\Delta L\downarrow_S$) and shortwave ($\Delta S\downarrow_S$) radiative fluxes at the surface, with respect to the base model run. Values (mW m$^{-2}$) are averages over 2006–2010.**

| Region | Parameter | Model difference from base run (mW m$^{-2}$) | | | | |
| --- | --- | --- | --- | --- | --- | --- |
| | | 1 (dep.) | 2 (dep. + LNO$_x$) | 3 (dep. + scaled LNO$_x$) | 4 (dep. + LNO$_x$ + no CH$_4$) | 5 (dep. + no LNO$_x$) |
| Globe (all sky) | $\Delta R_{TOA}^{N}$ | 4.4 | 86.3 | 70.9 | 107.0 | -190.8 |
| | $\Delta L_{TOA}^{N}$ | -2.5 | 74.0 | 54.8 | 101.2 | -184.3 |
| | $\Delta S_{TOA}^{N}$ | 6.9 | 12.3 | 16.1 | 5.8 | -6.5 |
| | $\Delta L\downarrow_S$ | 9.0 | 93.1 | 69.7 | 92.3 | -199.2 |
| | $\Delta S\downarrow_S$ | 7.8 | -72.1 | -44.6 | -75.6 | 204.9 |
| Tropics (all sky) | $\Delta R_{TOA}^{N}$ | 13.7 | 133.4 | 113.0 | 163.2 | -264.8 |
| | $\Delta L_{TOA}^{N}$ | 1.2 | 115.8 | 82.7 | 149.3 | -266.2 |
| | $\Delta S_{TOA}^{N}$ | 12.5 | 17.6 | 30.3 | 13.9 | 1.4 |
| | $\Delta L\downarrow_S$ | 4.7 | 131.0 | 90.5 | 135.1 | -283.7 |
| | $\Delta S\downarrow_S$ | 17.7 | -95.3 | -49.5 | -98.4 | 266.3 |
| Extra-tropics (all sky) | $\Delta R_{TOA}^{N}$ | -5.5 | 37.4 | 27.1 | 48.6 | -114.1 |
| | $\Delta L_{TOA}^{N}$ | -6.5 | 30.5 | 25.8 | 51.1 | -99.4 |
| | $\Delta S_{TOA}^{N}$ | 1.0 | 6.9 | 1.3 | -2.5 | -14.7 |
| | $\Delta L\downarrow_S$ | 13.4 | 53.8 | 48.2 | 47.9 | -111.4 |
| | $\Delta S\downarrow_S$ | -2.4 | -47.9 | -39.5 | -51.9 | 141.1 |
| Globe (clear sky) | $\Delta R_{TOA}^{N}$ | -6.0 | 110.8 | 77.6 | 132.9 | -276.5 |
| | $\Delta L_{TOA}^{N}$ | -7.4 | 95.2 | 69.2 | 123.0 | -245.2 |
| | $\Delta S_{TOA}^{N}$ | 1.4 | 15.6 | 8.4 | 9.9 | -31.3 |
| | $\Delta L\downarrow_S$ | 21.9 | 143.7 | 106.9 | 145.2 | -287.3 |
| | $\Delta S\downarrow_S$ | -0.2 | -70.0 | -54.3 | -72.8 | 177.4 |

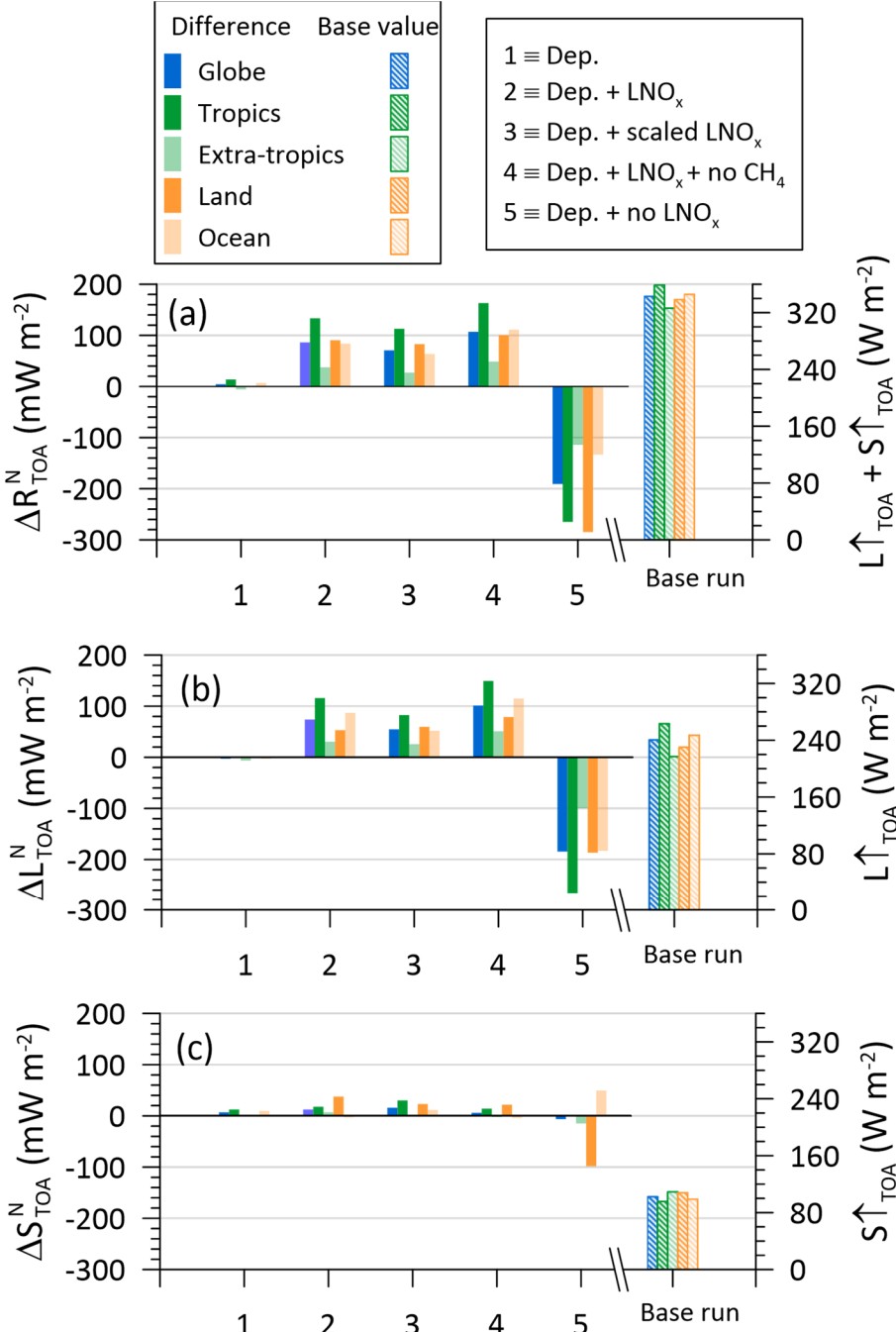

**Figure 2: Modelled mean (2006–2010) all-sky net downward TOA radiative flux difference between the various model runs and the base run (1 = Run B – Base, 2 = Run C – Base, 3 = Run D – Base, 4 = Run E – Base, 5 = Run F – Base) for the globe, tropics, extra-tropics, land, and ocean. The plots are for the (a) total radiative flux difference ($\Delta R_{TOA}^N$), (b) longwave radiative flux difference ($L_{TOA}^N$), and (c) shortwave radiative flux difference ($S_{TOA}^N$). The outgoing TOA flux values obtained from the base run are also plotted (corresponding to the right y-axis).**

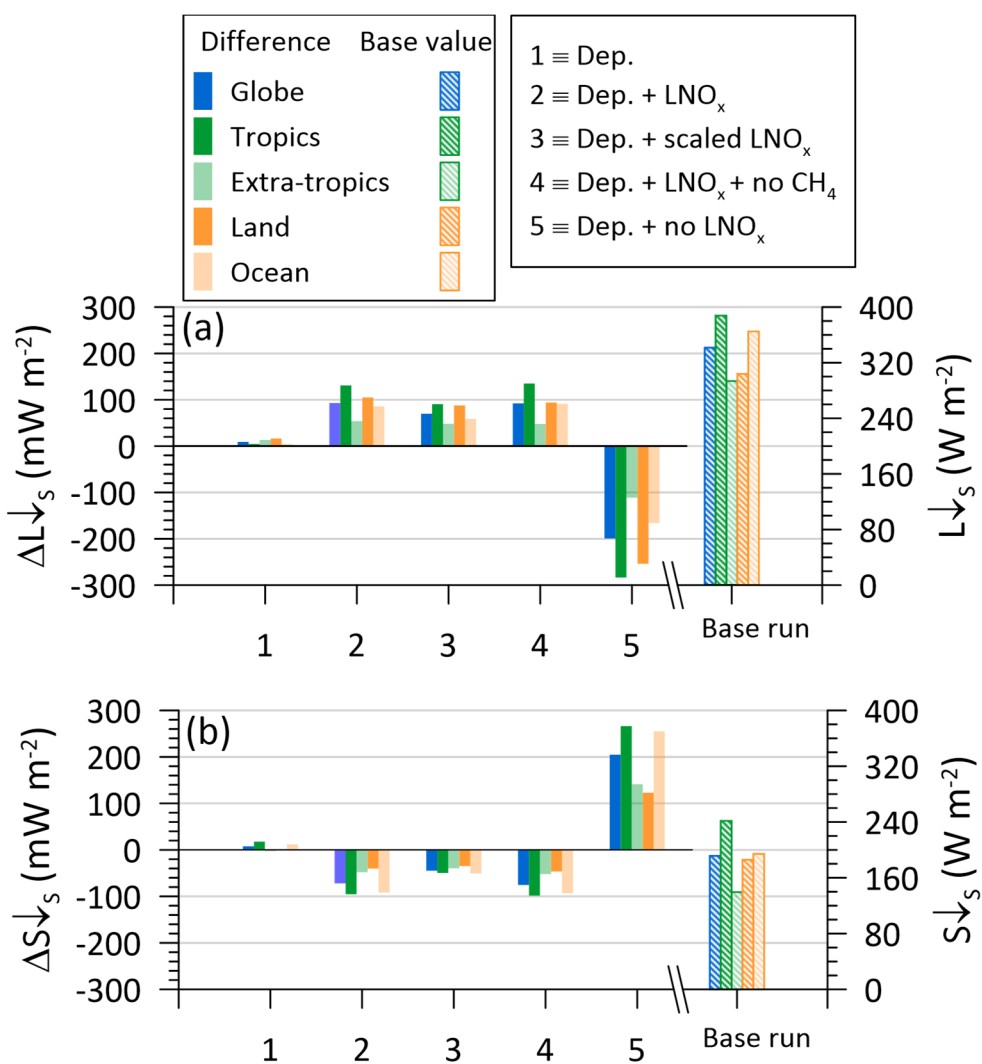

**Figure 3: Modelled mean (2006–2010) all-sky downward surface radiative flux difference between the various model runs and the base run (1 = Run B – Base, 2 = Run C – Base, 3 = Run D – Base, 4 = Run E – Base, 5 = Run F – Base) for the globe, tropics, extra-tropics, land, and sea. The plots are for the (a) longwave radiative flux difference ($\Delta L{\downarrow}_S$) and (b) shortwave radiative flux difference ($\Delta S{\downarrow}_S$). The downward surface flux ($L{\downarrow}_S + S{\downarrow}_S$) values obtained from the base run are also plotted (corresponding to the right y-axis).**

In Figure 4 (a, b and c), the variations of the modelled zonal means of $\Delta R^N_{TOA}$, $\Delta L\downarrow_S$ and $\Delta S\downarrow_S$ are presented (solid lines). The zonal means of the absolute total outgoing TOA radiative flux ($L\uparrow_{TOA} +S\uparrow_{TOA}$), $L\downarrow_S$ and $S\downarrow_S$ obtained from the base run are also plotted and compared with the corresponding CERES-EBAF data (dotted lines and solid circles, corresponding to the right y-axis).

In Figure 4a, the minimum in the modelled $L\uparrow_{TOA} +S\uparrow_{TOA}$ near the equator is mostly due to the high cloud tops associated with the inter-tropical convergence zone (ITCZ), which is a region of persistent thunderstorms, and the subtropical maxima are associated with clear air over deserts and subtropical highs. The radiative flux diminishes towards the poles, with the minimum being in the southern hemisphere polar region. There is a good agreement with the CERES-EBAF data, with some model overestimation between 50°N to 50°S and underestimation within 50–70°S. Looking at the differences $\Delta R^N_{TOA}$, dry deposition has little effect, but increased $LNO_x$ increases the flux from ~ 50°N to 30°S presumably due to increased emission of LW by $O_3$ produced by the $LNO_x$. Similarly, totally removing the $LNO_x$ decreases the TOA radiative flux particularly from approximately 30°S to virtually the north pole showing a marked hemispheric asymmetry towards the northern hemisphere. This contrasts, being in the opposite hemisphere, to the asymmetric effect of $LNO_x$ on the downward SW radiation at the surface (Figure 4c).

The radiative flux e $L\downarrow_S$ primarily depends on water vapour and temperature in the lower atmosphere and varies with increasing $CO_2$ and other greenhouse gases (Wang and Dickinson, 2013), including $O_3$ (Rap et al., 2015). In Figure 4b, $L\downarrow_S$ has a characteristic peak in the tropics and it diminishes poleward to a lower level in the northern polar region and to the lowest levels towards the southern hemisphere pole consistent with global climatologies (Wang and Dickinson, 2013). The model agreement with the CERES-EBAF data is excellent. In terms of $\Delta L\downarrow_S$, dry deposition has little effect, but increased $LNO_x$ increases the downward flux from ~ 40°N to 40°S presumably due to increased emission of LW by $O_3$ produced by the $LNO_x$. Similarly, totally removing the $LNO_x$ decreases LW radiation particularly from ~ 40°N to 40°S. This contrasts with the asymmetric effect of $LNO_x$ on SW radiation in Figure 4c.

The radiative flux $S\downarrow_S$ is affected by clouds which reflect and scatter solar radiation (see cloud climatology https://earthobservatory.nasa.gov/images/85843/cloudy-earth). In Figure 4c for $S\downarrow_S$, apparent are the characteristic peak from overhead solar radiation in the tropics, the influence of the tropical cloud band, the radiative flux diminishing to a low level in the northern polar region presumably due to widespread cloud cover there, and diminishing fluxes in the mid latitudes of the southern hemisphere and then increasing towards the pole consistent with cloud climatologies (https://earthobservatory.nasa.gov/images/85843/cloudy-earth). The model agreement with the CERES-EBAF data is good, with some model overestimation in the tropics. Considering the difference $\Delta S\downarrow_S$, dry deposition has little effect, but increased $LNO_x$ decreases the downward flux from ~ 20°N to 60°S. Similarly, totally removing the $LNO_x$ increases SW radiation from

~ 40°N to 70°S, which illustrates the asymmetric effect of $LNO_x$ across the hemispheres (and consequent asymmetric heating contribution). Generally, most of the SW radiation in the wavelength spectrum that $O_3$ can efficiently absorb is removed by the stratospheric $O_3$ such that little penetrates to the Earth's surface (Rap et al., 2015). Therefore, the decrease in $S\downarrow_S$ with $LNO_x$ is possibly not directly caused by the increased $O_3$ production as a result of the increased $LNO_x$, but could instead be driven by other factors such as changes in aerosol or cloud between the perturbed parameterisation experiments and the base run in response to changes in $LNO_x$. We have explored this in the next section.

Global distribution of the difference $\Delta R^N_{TOA}$ between the model Run C (i.e., the new dry deposition scheme and the new $LNO_x$ scheme; Diff. 2) and the base run (Figure 5a) is patchy with regions of both increased and decreased radiation compared to the base run, but with an overall increase. The difference in the incoming longwave radiative flux at the surface ($\Delta L\downarrow_S$) (Figure 5b) is positive almost everywhere over the globe, whereas that in the incoming shortwave radiative flux at the surface ($\Delta S\downarrow_S$) (Figure 5c) has a patchy global distribution with regions of both positive and negative values. The respective area-weighted global spatial means of these differences in Figure 5a, b and c are $86.3 \pm 387.3$, $93.1 \pm 184.1$ and $-72.1 \pm 588.9$ mW m$^{-2}$, where the standard deviations were obtained from area-weighted variances and their relatively large values reflect the spatial heterogeneity of the radiation response.

Figure 5d presents the corresponding difference in lightning flash density ($\Delta f$) between the two models, which shows a larger flash density predicted everywhere by the new $LNO_x$ scheme, particularly over the ocean in the tropics ($LNO_x$ is directly proportional to the lightning flash rate in our model). The area-weighted spatial pattern correlation ($r$) between $\Delta f$ and $\Delta R^N_{TOA}$ is 0.15. The correlation of $\Delta f$ with $\Delta L\downarrow_S$ is 0.33 and with $\Delta S\downarrow_S$ it is -0.14. These relatively low correlations imply that while the $LNO_x$ production occurs (and changes from model run to run) in one spatial pattern, time is required for chemical processing from $NO_x$ to $O_3$ and $CH_4$ and during this time advection and dispersion take place, and also the feedbacks from the impact on aerosol and cloud cover, so that the radiative effects occur in a different spatial pattern compared to the lightning flash rate. The area-weighted spatial pattern correlations between the difference in the ozone column ($\Delta DU$) in Figure 1, and $\Delta R^N_{TOA}$, $\Delta L\downarrow_S$ and $\Delta S\downarrow_S$ are 0.24, 0.48 and -0.14, respectively.

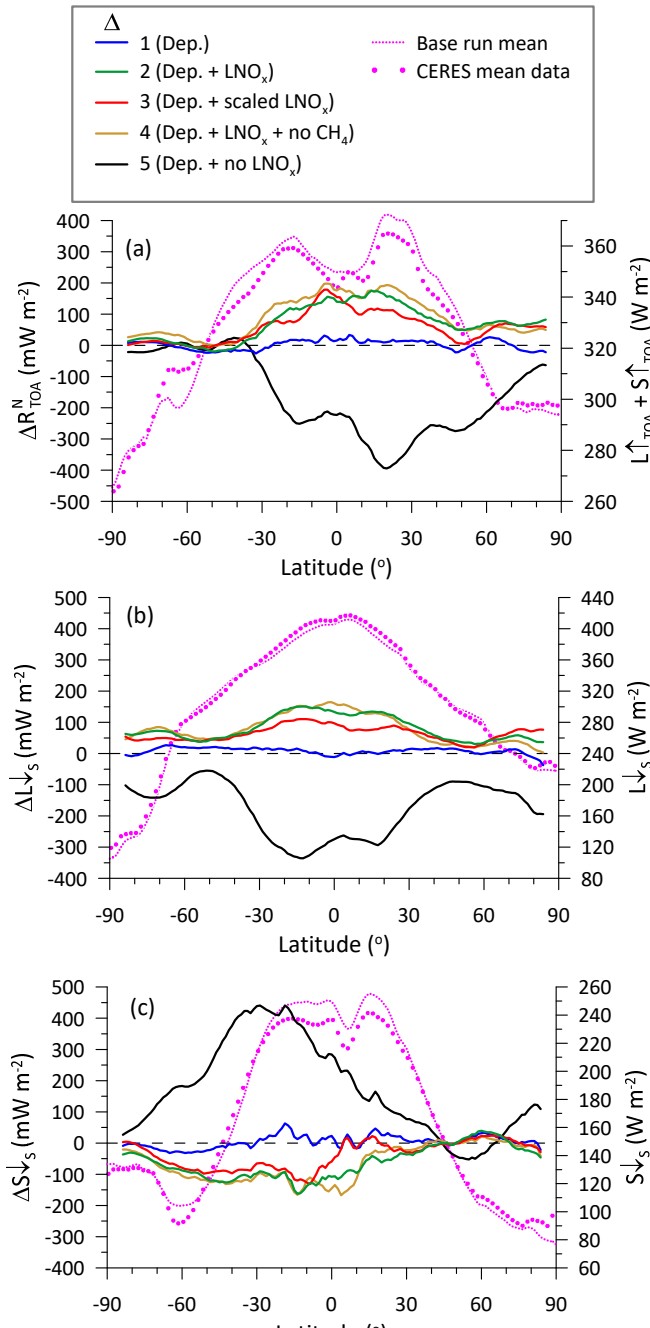

**Figure 4: Zonal mean (2006–2010) of all-sky radiative flux difference between the various model runs and the base run (1 = Run B – Base, 2 = Run C – Base, 3 = Run D – Base, 4 = Run E – Base, 5 = Run F – Base). The plots (solid lines) are for the (a) net downward TOA radiative flux difference ($\Delta R^N_{TOA}$), (b) downward longwave radiative flux difference at the surface ($\Delta L\downarrow_S$) and (b) downward shortwave radiative flux difference at the surface ($\Delta S\downarrow_S$). The solid lines are running averages (over a moving window of 10 points, i.e., 12.5°). The zonal mean flux values obtained from the base run and the corresponding CERES-EBAF data are also plotted (dotted line and solid circles, corresponding to the right y-axis).**

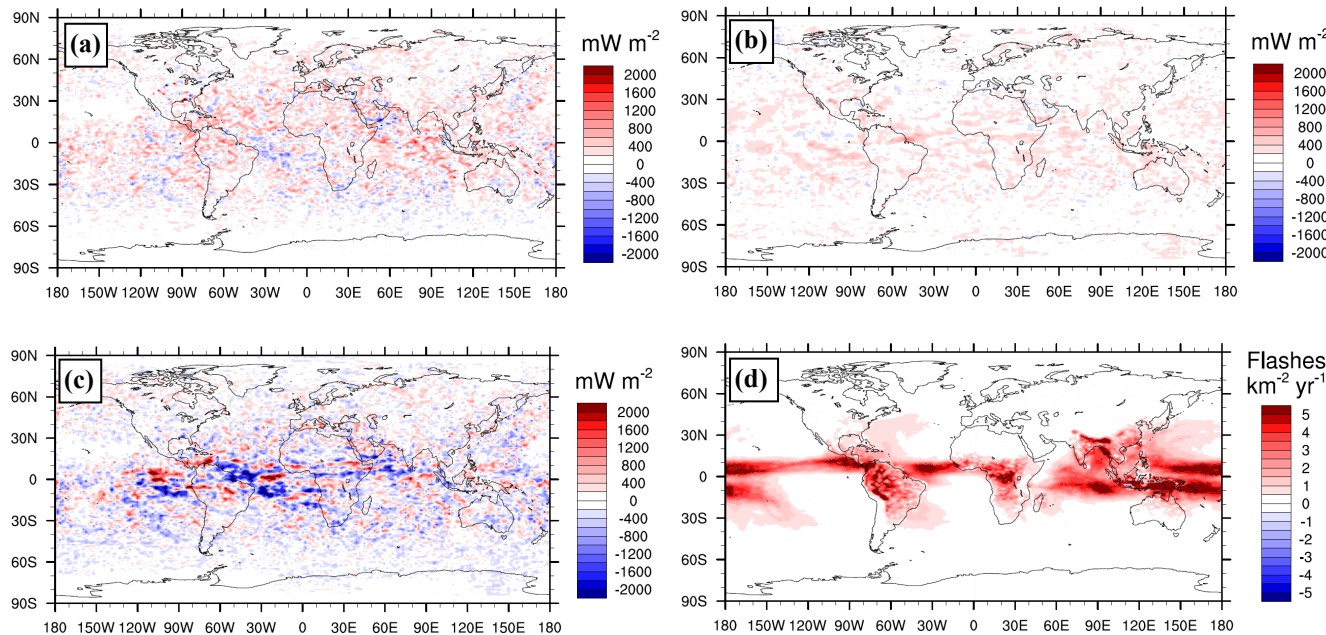

**Figure 5: Global distribution of the all-sky radiative flux difference (mW m$^{-2}$) between Run C (i.e., the new dry deposition scheme and the new LNO$_x$ scheme) and the base run (mean over 2006–2010). The plots are for the (a) net downward TOA radiative flux difference ($\Delta R_{TOA}^{N}$), (b) incoming longwave radiative flux difference at the surface ($\Delta L\downarrow_S$) and (c) incoming shortwave radiative flux difference at the surface ($\Delta S\downarrow_S$). In (d), the corresponding model difference in lightning flash density, given as flashes km$^{-2}$ yr$^{-1}$, is shown.**

## 3.4 Changes in incoming surface shortwave radiation, aerosol and cloud cover

As stated earlier, the decrease in all-sky $S\downarrow_S$ cannot not be explained by the increased O$_3$ production as LNO$_x$ is increased, and therefore to further understand what drives the differences in the shortwave flux at the surface we look at any changes in aerosol fields and cloud cover that may explain this decrease. We only consider the parameter value differences between the model run with both the new oceanic O$_3$ dry deposition scheme and the new lightning flash-rate parameterisation (Run C) and the base model run, and the zonal means of these differences are shown in Figure 6. The increased LNO$_x$ in Run C also leads to a decrease in the clear-sky $\Delta S\downarrow_S$ and this corresponds to an increase in the column integrated CN (or aerosol) number concentration. Increases in the column integrated CN by as much as $2 \times 10^9$ cm$^{-2}$ are found. These aerosol in addition to reflecting and scattering solar radiation, also affect clouds. The all-sky $\Delta S\downarrow_S$ is more asymmetric across the hemispheres than the clear-sky $\Delta S\downarrow_S$, which could be due to the hemispheric asymmetric in $\Delta$CN coupled with influence of changes in cloud properties.

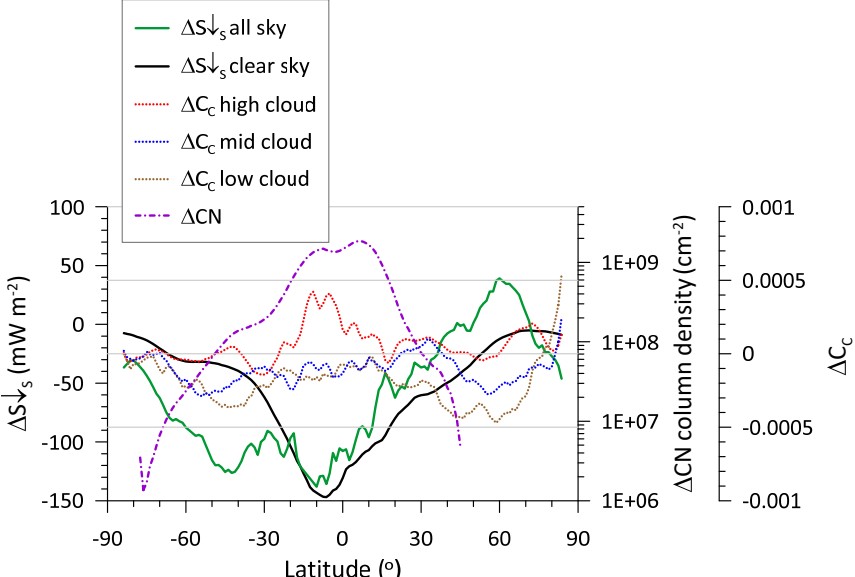

**Figure 6: Zonal mean (2006–2010) of parameter value differences between the model run with both the new oceanic O₃ dry deposition scheme and the new lightning flash-rate parameterisation and the base model run (Run C – Base): all-sky downward shortwave radiative flux difference at the surface (ΔS↓ₛ); clear-sky ΔS↓ₛ; differences in high, medium and low cloud cover (ΔC꜀); and column integrated condensation nuclei (CN) concentration difference (ΔCN). The lines are running averages (over a moving window of 10 points, i.e., 12.5°).**

We also calculated changes in fractional cloud cover $C_C$ (which varies between 0–1) for low (< 2 km), middle (2–6 km) and high (> 6 km) level clouds. Cloud cover is important for the modelling of downward radiation (Chen et al., 2012), and the model output for this quantity was available for each grid box (volume fraction of a gridbox covered in cloud) at each model level. The total cloud cover within the above three cloud-height categories can be calculated approximately from the modelled cloud cover at each model layer using a cloud overlap assumption. We used the combined maximum/random cloud overlap assumption which lies between the random overlap assumption (which overestimates the total cloud cover) and the maximum overlap assumption (which underestimates the cloud cover) (Oreopoulos and Khairoutdinov, 2003). Figure 6 shows the zonal means of cloud cover differences (Δ$C_C$) between Run C and the base model run for the high, middle, and low level clouds. It is apparent that cloud cover is impacted at all levels by the model changes, with more high-level and less middle- and low-level cloud cover, although the degree of changes being small, all within ± 0.05%. The high-level zonal mean Δ$C_C$ appears to be noticeably anti-correlated with all-sky zonal mean ΔS↓ₛ in the tropics whereas for northern latitudes above about 40° both middle- and low-level zonal mean Δ$C_C$ are anti-correlated with ΔS↓ₛ.

Figure 7 presents the global spatial distributions of Δ$C_C$ for the three cloud-height categories, which are patchy with regions of both positive and negative values. But visually comparing Figure 7a for the high-level cloud cover with the all-sky ΔS↓ₛ

global distribution in Figure 5c, one can clearly notice that regions of $\Delta S\downarrow_S$ are anticorrelated with $\Delta C_C$, and that this anticorrelation becomes progressively weaker for middle- and low-level $\Delta C_C$ in Figure 7b and Figure 7c, respectively. The area-weighted spatial pattern correlation between $\Delta S\downarrow_S$ and $\Delta C_C$ is -0.44, -0.39, and -0.30 for the high-, middle-, and low-level cloud, respectively.

The above results suggest that while the decrease in the all-sky $\Delta S\downarrow_S$ with increased $LNO_x$ may not possibly be explained in terms of the consequent $O_3$ production, the indirect effect of $LNO_x$ on aerosol and cloud can at least partly explain the differences in the shortwave flux at the surface.

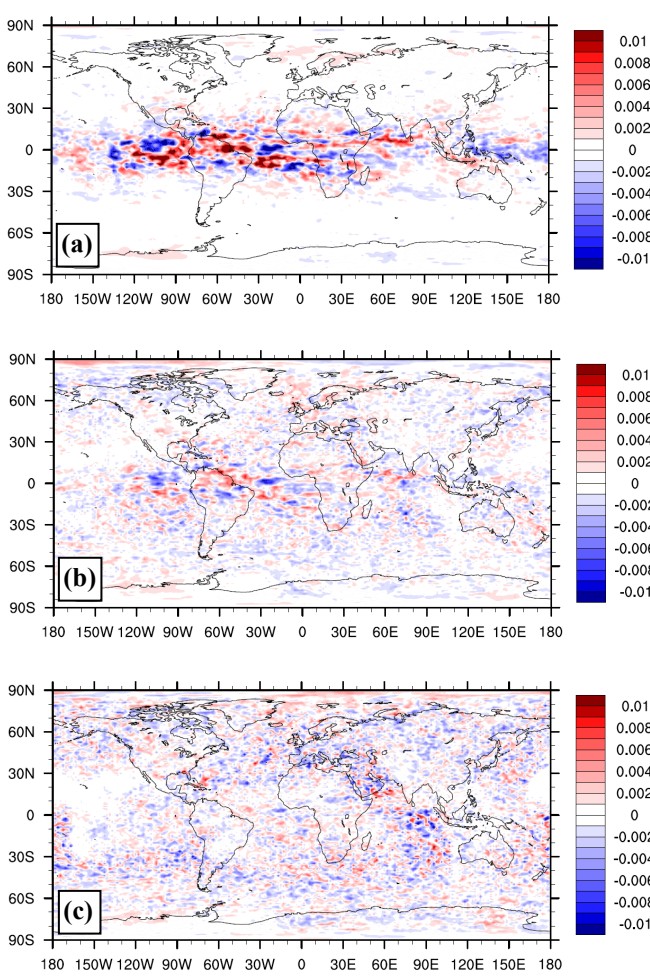

**Figure 7: Global distribution of the modelled fractional cloud cover difference (($\Delta C_C$) between Run C (i.e., the new dry deposition scheme and the new $LNO_x$ scheme) and the base run (mean over 2006–2010). The plots are for (a) high level (> 6 km), (b) middle level (2–6 km) and (c) low level (< 2 km) clouds.**

### 3.5 Radiative effects as a function of changes in LNO$_x$

Lightning NO$_x$ production is a natural process with effects on the atmospheric radiation and energy budget via O$_3$, CH$_4$, and aerosol. Lightning NO$_x$ is different from the bulk of other natural and anthropogenic sources of NO$_x$ in the troposphere in that the release occurs in the upper portion of the troposphere rather than close to the Earth's surface.

In Figure 8a, we plot the change in the modelled net downward total TOA radiation ($\Delta R_{TOA}^{N}$) as a function of change in the annual-averaged LNO$_x$ production relative to the base model run. The different points are for different years and different model runs. All runs except Run E (i.e., without the CH$_4$ radiation feedback) are considered. The plot shows an approximately linear increasing change in the net downward total TOA radiation as a function of increase in the LNO$_x$ production due to the various model parameterisation/configuration changes considered. (There is a relatively large gap between the LNO$_x$ = 0 case and the LNO$_x$ = 4.8 Tg N yr$^{-1}$ case. To confirm that the linear fit is not unduly dominated by the LNO$_x$ = 0 case and that the linearity is appropriate, an additional model simulation the same as Run C but with the LNO$_x$ distribution scaled uniformly by 0.35 to give an averaged total LNO$_x$ = 2.4 Tg N yr$^{-1}$ was made, and the results from this simulation are also plotted in Figure 8–Figure 9 enclosed by a dotted circle and they are included in determining the linear fits.)

The slope of the best fit lines in Figure 8a suggests that with a per Tg increase in N production per year due to lightning, there is an increase of 39.6 mW m$^{-2}$ (Tg N yr$^{-1}$)$^{-1}$ in the net downward TOA radiation, or that much of radiation is retained by the atmosphere. Similarly, based on the slopes of the best fit lines in Figure 8b and Figure 8c, there is an increase of 40.2 mW m$^{-2}$ (Tg N yr$^{-1}$)$^{-1}$ in the incoming surface longwave radiation, and a decrease of 36.4 mW m$^{-2}$ (Tg N yr$^{-1}$)$^{-1}$ in the incoming surface shortwave radiation.

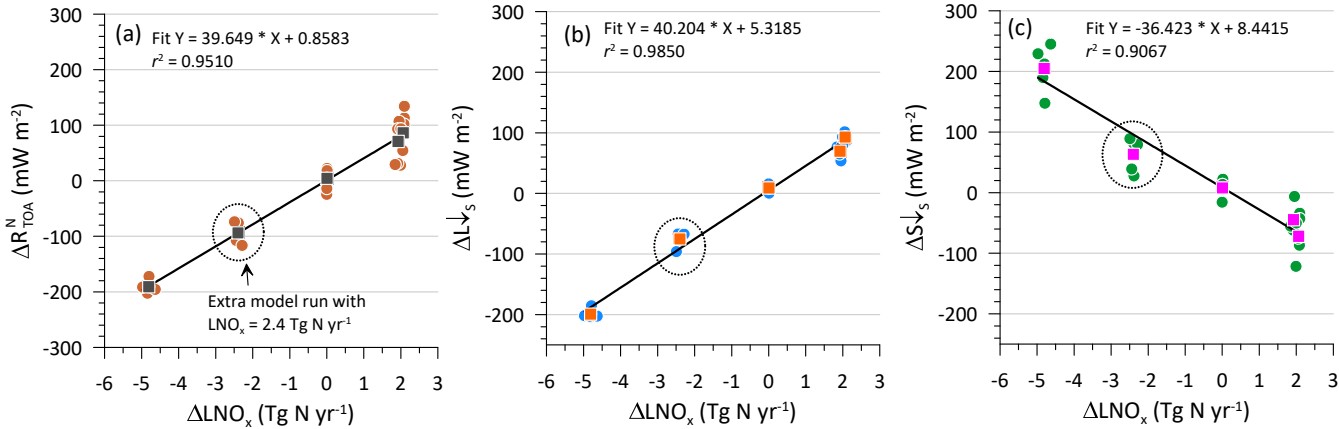

**Figure 8: Change in the modelled (a) net downward total TOA radiation ($\Delta R^N_{TOA}$), (b) incoming surface longwave radiation ($\Delta L{\downarrow}_S$), and (c) incoming surface shortwave radiation ($\Delta S{\downarrow}_S$), as a function of change in the modelled lightning-generated $NO_x$ ($\Delta LNO_x$, Tg N per year) relative to the Base model run. All runs except Run E (i.e., without the $CH_4$ radiation feedback) are considered. Results from an additional model simulation the same as Run C but with averaged $LNO_x = 2.4$ Tg N $yr^{-1}$ are also plotted (enclosed in a dotted circle) to check linearity. The solid circles are the annual means whereas the squares are the mean over 2006–2010. The best fit line is based on all points.**

### 3.6 Radiative effects as a function of changes in column O₃

Radiation is examined with respect to $O_3$ column changes (Figure 9) caused by the dry deposition and $LNO_x$ parameterisation changes. Based on Figure 9a, with a per DU increase in $O_3$, there is an increase of 22.8 mW m$^{-2}$ DU$^{-1}$ in the net downward TOA radiation, when the $O_3$ change is dominated by an increase in $LNO_x$. This can be compared to normalised radiative forcing calculations reported in the scientific literature. Using the results of 17 atmospheric chemistry models, Stevenson et al. (2013) derived a globally averaged normalized radiative forcing of 42 (range 36–45) mW m$^{-2}$ DU$^{-1}$ for tropospheric $O_3$ increase from preindustrial (1750) to present day (2010). Gauss et al. (2003) calculated a normalized radiative forcing of $36 \pm 3$ mW m$^{-2}$ DU$^{-1}$ due to changes in tropospheric $O_3$ between 2000 and 2100 based on the results of 11 models. In our study, the changes in TOA radiative effects per DU, which are primarily driven by changes in the $LNO_x$ parameterisation and to a much lesser extend by the $O_3$ deposition parameterisation (with $CH_4$ feedbacks included), are about half the radiative forcing per DU as a result of climate-scale changes in tropospheric $O_3$ (e.g., due to changes in precursor emissions and temperature). The climate-scale changes in radiation due to $O_3$ are larger, possibly because there are co-emissions of non-$NO_x$ precursor species (e.g., $CH_4$ and VOCs) and their feedbacks, whereas in the present case only $LNO_x$ and $O_3$ dry deposition changes are considered (together with $CH_4$ feedbacks). Methane levels and VOC emissions are unchanged. In any event, a clear distinction should be made between the expected column ozone changes arising from a change in $LNO_x$ emissions versus a change in anthropogenic $NO_x$ emissions.

Similarly, based on Figure 9b and Figure 9c, there is an increase in downward surface longwave radiation $L\downarrow_S$ by 23.6 mW m$^{-2}$ DU$^{-1}$ and a decrease in downward surface shortwave radiation $S\downarrow_S$ by 21.7 mW m$^{-2}$ DU$^{-1}$. (As discussed earlier, changes in $S\downarrow_S$ can be explained in part by changes in aerosol fields and cloud cover as $LNO_x$ is increased, but it is clear here that there is a good statistical correlation between $\Delta S\downarrow_S$ and $\Delta DU$).

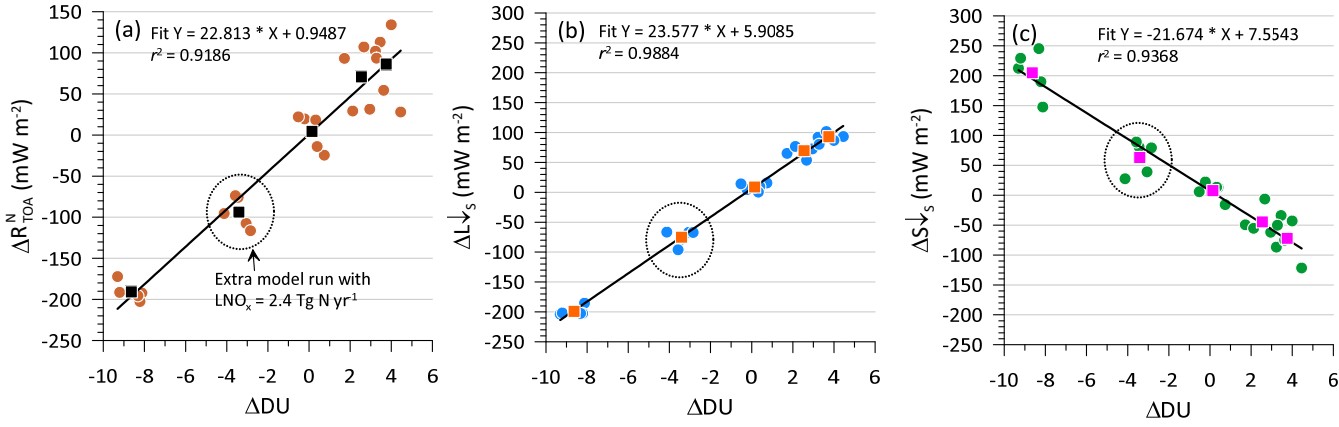

**Figure 9: Change in the modelled (a) net downward total TOA radiation ($\Delta R^N_{TOA}$), (b) incoming surface longwave radiation ($\Delta L\downarrow_S$) and (c) incoming surface shortwave radiation ($\Delta S\downarrow_S$), as a function of change in the modelled O$_3$ column ($\Delta DU$) relative to the Base model run. All runs except Run E (i.e., without the CH$_4$ radiation feedback) are considered. Results from an additional model**
**simulation the same as Run C but with averaged $LNO_x$ = 2.4 Tg N yr$^{-1}$ are also plotted (enclosed in a dotted circle) to check linearity. The solid circles are the annual means whereas the squares are the mean over 2006–2010. The best fit line is based on all points.**

The above changes in radiation were also examined as a function of changes in tropospheric O$_3$ burden (plots not shown). The slopes of these plots indicate that there is an increase of 3.3 mW m$^{-2}$ (Tg O$_3$)$^{-1}$ in the net downward TOA radiation, an increase
of 3.3 mW m$^{-2}$ (Tg O$_3$)$^{-1}$ in the incoming surface longwave radiation, and a decrease of 3.1 mW m$^{-2}$ (Tg O$_3$)$^{-1}$ in the incoming surface shortwave radiation.

In this paper, we have not explored atmospheric temperature response or any changes in atmospheric heating rates caused by the changes in the radiation balance due to the use of the improved parameterisations.

### 3.7 LNO$_x$ and the tropospheric lifetime of CH$_4$

In Figure 10, the change in $\tau_{CH_4}$ plotted as a function of change in $LNO_x$ suggests that there is a shortening of the global mean CH$_4$ lifetime by 0.31 years per Tg N yr$^{-1}$ produced due to lightning. This change in $\tau_{CH_4}$ is equivalent to a change of –4.4% $\tau_{CH_4}$ per Tg N yr$^{-1}$ produced due to lightning (with respect to the Run C value of $\tau_{CH_4}$), which is close to the average –4.8 (range –6.8 to –2.4) % $\tau_{CH_4}$ per Tg N yr$^{-1}$ given by Thornhill et al. (2021a) based on four other models. The uncertainty in

$\tau_{CH_4}$ corresponding to an LNO$_x$ uncertainty range of $5 \pm 3$ Tg N yr$^{-1}$ (Schumann and Huntrieser, 2007) would thus be $\pm 0.92$ years or $\pm 13.5\%$ of $\tau_{CH_4}$ from Run C.

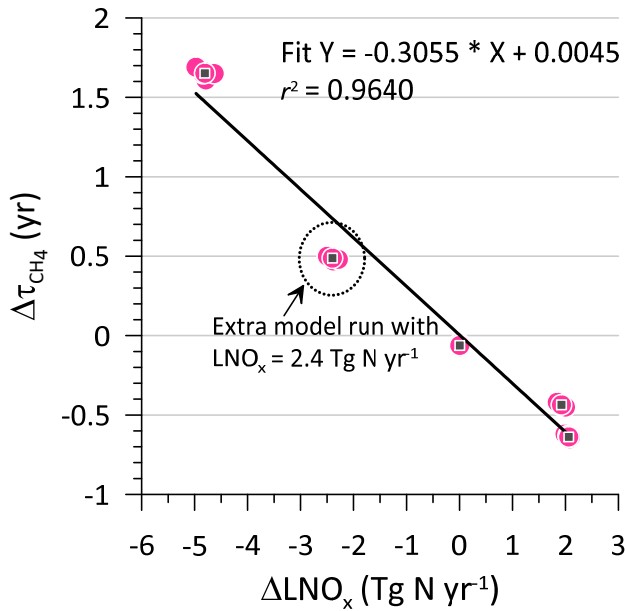

5 **Figure 10: Same as Figure 8, except for change in the modelled CH$_4$ lifetime ($\Delta\tau_{CH_4}$).**

### 3.8 LNO$_x$ and column integrated CN concentration

The change in in the modelled global column integrated CN concentration is plotted as a function of change in LNO$_x$ in Figure 11. The slope of the linear fit suggests that there is an increase in the column CN concentration by $0.163 \times 10^9$ cm$^{-2}$ per Tg N
10  yr$^{-1}$ LNO$_x$. Thus, the uncertainty in the column CN concentration corresponding to an LNO$_x$ uncertainty range of $5 \pm 3$ Tg N yr$^{-1}$ would be $\pm 0.49 \times 10^9$ cm$^{-2}$ or $\pm 6.2\%$ of the column CN concentration obtained from Run C.

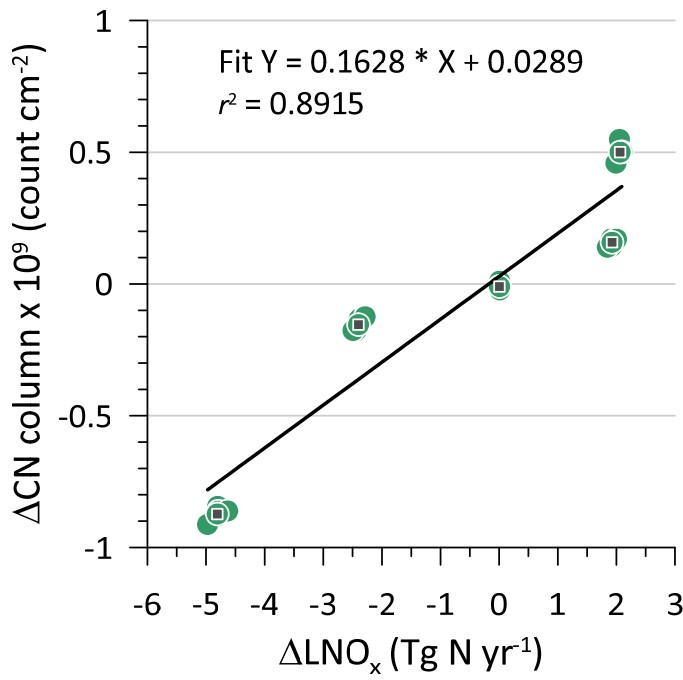

Fit Y = 0.1628 * X + 0.0289
$r^2$ = 0.8915

**Figure 11: Same as Figure 8, except for change in the modelled global column integrated CN concentration.**

### 3.9 LNO$_x$ and radiative forcing in a broader context

The magnitude of the modelled net downward TOA radiation differences ($\Delta R_{TOA}^{N}$) obtained (in section 3.3) can be put in the context of the IPCC AR6 reported anthropogenic ERF due to O$_3$ over the years 1750–2019 of $470 \pm 230$ mW m$^{-2}$ (Forster et al., 2021), noting that the radiation differences calculated here are akin to instantaneous radiative forcing, which excludes any adjustments (e.g., adjustments causing circulation changes) unlike the ERF which is the sum of the IRF and the contribution from the adjustments. For example, the extra TOA radiation of 86.3 mW m$^{-2}$ retained by the atmosphere when the new deposition and lightning flash-rate schemes are used, which represents the uncertainty in radiation due to two natural process representation in chemistry-climate models, is equivalent to 18% of the IPCC AR6 reported anthropogenic O$_3$ ERF of 470 mW m$^{-2}$.

Using the amount of radiative flux change per Tg N change in LNO$_x$ from section 3.5 and assuming an uncertainty range of $5 \pm 3$ Tg N yr$^{-1}$ in the global estimates of LNO$_x$ suggested by Schumann and Huntrieser (2007), the corresponding uncertainty range in the net downward TOA radiation retained in the atmosphere could thus be as much as $\pm 119$ mW m$^{-2}$. (Although these cannot be compared directly, this is equivalent to 50% of the IPCC AR6 reported anthropogenic O$_3$ ERF.) Similarly, the corresponding uncertainty range is $\pm 121$ mW m$^{-2}$ for the surface longwave radiation and $\pm 109$ mW m$^{-2}$ for the surface shortwave radiation. Thus, the implications of this uncertainty in LNO$_x$ for global climate modelling needs investigation and clarification.

As demonstrated in this study, the net instantaneous radiative forcing of $LNO_x$ is positive with the enhanced $O_3$ production dominating over the reduced $CH_4$ lifetime. The emission-based ERF due to increases in anthropogenic $NO_x$ emissions (from 1750 to 2019) based on chemistry-climate models is reported to be negative (at $-0.29 \pm 0.29$ W m$^{-2}$), which is a net effect of a positive ERF through enhanced tropospheric $O_3$ production, a negative ERF through reduced $CH_4$ lifetime, and a small negative ERF contribution through formation of nitrate aerosols (Naik et al., 2021). Notwithstanding the differences between ERF and IRF, this contrast between the $LNO_x$ and anthropogenic $NO_x$ forcings could at least be partially because (a) in the upper to middle troposphere within which $LNO_x$ is generated, the production efficiency of $O_3$ per unit of $NO_x$ is much larger than that close to the Earth's surface, where anthropogenic emissions are mostly released (Dahlmann et al., 2011) and (b) the historical anthropogenic $NO_x$ emissions were accompanied by emissions of reactive VOCs which affects the subsequent chemistry.

As shown in section 3.7, $LNO_x$ has a significant influence on the atmospheric lifetime of $CH_4$, and the value of $LNO_x$ used within a model will influence the time integrated measures of radiative forcing including the ERF and the global warming potential (GWP) of anthropogenic $CH_4$.

Recent chemistry-climate modelling studies have explored changes in $LNO_x$ in a future warming climate but there remains a large uncertainty depending on how lightning flash-rate parameterisations are formulated. All CMIP6 Earth system models use flash-rate parameterizations that use convective cloud-top height (as is the case in the present paper) and they project an increase in lightning and hence in $LNO_x$ in a warmer world of 0.27–0.61 Tg N yr$^{-1}$ per °C (Naik et al., 2021; Thornhill et al., 2021a). Flux-based flash-rate parameterisations, e.g., that by Finney et al. (2018) using upward cloud ice flux, predict decrease in lightning under climate change. Thus, despite the improvements in understanding, $LNO_x$ remains a significant uncertainty for climate and earth system modelling.

**4 Conclusions**

The impact of recent process-based improvements to oceanic $O_3$ dry deposition parameterisation (Luhar et al., 2018) and empirical improvements to lightning-generated $NO_x$ parameterisation (Luhar et al., 2021) on radiative transfer was investigated via the use of the ACCESS-UKCA chemistry-climate model, which includes radiative feedbacks of $O_3$, $CH_4$ and aerosol. The main radiation components examined were the net downward TOA radiative flux, and the incoming longwave and shortwave radiation at the Earth's surface.

The effects of the $LNO_x$ parameterisation change (which enhanced the $LNO_x$ production from 4.8 to 6.9 Tg N yr$^{-1}$) were a factor of roughly 10 to 20 larger than those due to the dry deposition change. The two combined parameterisation changes increased the global tropospheric $O_3$ burden by 31.9 Tg $O_3$ (11.7%), increased the global $O_3$ column by 3.75 DU (13% of the tropospheric column or 1.2% of the total column), decreased the global mean tropospheric lifetime of $CH_4$ by 0.64 years (8.4%), increased the global column integrated aerosol number concentration by $0.5 \times 10^9$ cm$^{-2}$ (6.7%), and impacted the cloud cover somewhat (zonal mean value by as much as $\pm 0.05\%$).

The use of the improved oceanic dry deposition scheme resulted in a relatively small increase of 4.4 mW m$^{-2}$ in the globally-averaged all-sky net downward TOA radiative flux (i.e., that much of radiation is retained by the atmosphere), but this increase was much larger at 86.3 mW m$^{-2}$, most of which longwave, when the improved LNO$_x$ parameterisation was also used (this increase was 107.0 mW m$^{-2}$ when the CH$_4$ radiative feedback was neglected). This change in the radiative flux represents a measure of uncertainty in radiation due to two natural processes represented in chemistry-climate models, and, for comparison, is equivalent to 18% of the IPCC AR6 reported present-day anthropogenic radiative forcing due to O$_3$ of 470 mW m$^{-2}$.

Similarly, with the two parameterisation changes, there was an increase of 93 mW m$^{-2}$ in the all-sky downward longwave radiation and a decrease of 72 mW m$^{-2}$ in the all-sky downward shortwave radiation at the Earth's surface. The changes in the all-sky downward shortwave radiation at the surface were consistent with the changes in the column aerosol number concentration and high-level cloud cover in response to the parameterisation changes.

The radiation changes due to the two improved parameterisations were larger in magnitude in the tropics than elsewhere. It was also found that when the default PR92 lightning flash-rate scheme (which underestimates the flash-rate distribution considerably over the ocean) was used with a uniform global scaling so as to give the total global LNO$_x$ the same as the improved scheme, the improved scheme yielded a larger net downward TOA radiation by ~15 mW m$^{-2}$, which implies that how LNO$_x$ is distributed spatially makes a difference to how the radiative transfer is impacted.

Based on the slopes of linear fits, with a per Tg N yr$^{-1}$ increase in LNO$_x$, there was an increase of 39.6 mW m$^{-2}$ in the net downward TOA radiation, an increase of 40.2 mW m$^{-2}$ in the incoming surface longwave radiation, a decrease of 36.4 mW m$^{-2}$ in the incoming surface shortwave radiation, and a shortening of CH$_4$ lifetime by 0.31 years (or ~ 4%).

The uncertainty range in the all-sky net downward TOA radiative flux due to reported uncertainty range of $5 \pm 3$ Tg N yr$^{-1}$ in global estimates of LNO$_x$ could be as much as $\pm 119$ mW m$^{-2}$. This value is equivalent to 50% of the present-day anthropogenic radiative forcing due to O$_3$ reported by the IPCC AR6. The corresponding modelled uncertainty range is $\pm 121$ mW m$^{-2}$ for the surface longwave radiation, $\pm 109$ mW m$^{-2}$ for the surface shortwave radiation, and $\pm 0.92$ years for CH$_4$ lifetime.

The above results highlight the impact of LNO$_x$ on tropospheric O$_3$ production, methane lifetime and aerosol, with ramifications for the Earth's radiation budget, and suggest that the value of LNO$_x$ used within a model will influence the modelled ERF and GWP of anthropogenic methane. The inter-model uncertainty in the ERF for methane will be contributed to by the model choices of LNO$_x$.

**Data availability**

The global model output (in NetCDF) from the ACCESS-UKCA simulations made can be obtained by contacting the corresponding author (Ashok Luhar: ashok.luhar@csiro.au). Radiation observations used for model comparison purposes were

available from NASA's Clouds and the Earth's Radiant Energy System (CERES) EBAF (Energy Balanced and Filled) Ed4.1 dataset (https://ceres.larc.nasa.gov/data/).

**Author contributions**

All authors had the initial idea to carry out this work. AKL designed the study, performed the model simulations, analysed the model output and data, and prepared the manuscript. IEG co-designed the study, advised on various components of the paper, and contributed to paper writing. MTW advised on the model setup. All authors contributed to revisions of the paper.

**Competing interests**

The authors declare that they have no conflict of interest.

**Acknowledgements**

This work was partly supported by the Climate Systems Hub of the National Environmental Science Program (NESP) funded by the Australian Government, and was undertaken with the assistance of resources and services from the National Computational Infrastructure (NCI), which is supported by the Australian government. ERA-Interim data from the European Centre for Medium-Range Weather Forecasts (ECMWF) and radiation data from NASA's Clouds and the Earth's Radiant Energy System (CERES) were used in this research. We acknowledge Martin Dix of CSIRO for his help with model

configuration issues, and Luke Abraham of the University of Cambridge and Mohit Dalvi of the U.K. Met Office for answering technical questions on the UM-UKCA model setup. We thank the two anonymous reviewers for their constructive comments.

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
