# Peer review of "Radiative impact of improved global parameterisations of oceanic dry deposition of ozone and lightning-generated $NO_x$"

_Atmospheric Chemistry and Physics, 2022_

## Author Comment (AC1)

**Reply by the authors to Referee #1's comments on**
**"Radiative impact of improved global parameterisations of oceanic dry deposition of ozone and lightning-generated NO$_x$" (#acp-2022-275)**

**Anonymous Referee #1 (RC1)**

We are grateful to the Referee for taking the time to review our manuscript and making a number of valuable comments, which have significantly improved the quality of the manuscript. In the following, we provide our responses to these comments (the Referee's comments are shown in blue). The locations of the changes made refer to those in the non-tracked version of the revised manuscript.

This paper presents an analysis of the radiative impact associated with changes to two model parameterisations: oceanic ozone deposition and production of lightning-generated NOx. The changes to the two parameterisation schemes and the impact on ozone distributions are described in previous papers while in this work the authors focus on a number of model experiments designed to evaluate changes in radiative fluxes and attribute them to changes in ozone and methane.

The topic of the paper and the methods used are sound and make it suitable for publication. In particular, the authors' findings that uncertainty in LNO emissions can have a large impact on global climate modelling has wide implications in the field. However, there is not a lot of analysis on the factors driving the changes in radiative fluxes.

One main concern is on the modelled changes to the shortwave radiation fluxes at the surface. This is an interesting and unexpected result, and therefore grants further investigation as it is not clear what causes this reduction when LiNOx is increased (p15, l13-17 and Fig 2b, 3c and 4c). This is unlikely to be due to increased absorption by increased ozone concentrations (as suggested by the authors) because this is inconsistent with latitudes where changes in downward longwave radiation at the surface (fig 3b) due to increased ozone are largest. Also, most of the shortwave radiation in the wavelength spectrum that ozone can efficiently absorb is already removed by stratospheric ozone (with tropospheric ozone only accounting for 1-3 mW/m2 in the shortwave (see Rap et al. 2015). Based on what is shown here (including fig 4c), it seems to me that whilst changes in the longwave are consistent with increased ozone production, as suggested by the authors, changes in the shortwave could instead be driven by some other factors, possibly including some changes in cloud or aerosols between the perturbed parameterisation experiments and the base run. I suggest the authors look at differences in the cloud and aerosols fields between various runs and the base run to further understand what drives differences in the shortwave fluxes at the surface.

**Response:** We thank the referee for making this point. To further investigate why the incoming shortwave (SW) radiation flux at the surface is reduced given that this behaviour cannot possibly be explained in terms of increased absorption by increased ozone concentrations in response to an increase in LNO$_x$, we have now analysed changes in the SW flux at the surface vis-à-vis changes in the modelled condensation nuclei (CN) (or aerosol number) concentration and cloud cover. We considered the parameter changes between the model run with both the improved oceanic O$_3$ dry deposition scheme and the improved lightning flash-rate parameterisation (Run C), and the base

model run. This analysis is reported in a new section 3.4 entitled "Changes in incoming surface shortwave radiation, aerosol and cloud cover" with new plots (Figures 1, 6, 7 and 11). The overall conclusion, as suspected by the Referee, is that the aerosol and cloud cover are impacted by changes in $LNO_x$ and that the modelled changes in high-level cloud cover can at least partly explain the differences in the shortwave fluxes at the surface.

One could also possibly examine other modelled parameters such as the cloud condensation nuclei (CCN) concentration or aerosol optical depth (AOD) but the model output for these quantities was not available. We think that the CN and cloud cover analysis reported in the revised paper should suffice to demonstrate that aerosol/cloud can, to some degree, explain the decrease in the SW radiation at the surface with an increase in $LNO_x$.

The paper by Rap et al. (2015) is now included and commented upon.

**Changes in manuscript:** As above. New section 3.4 "Changes in incoming surface shortwave radiation, aerosol and cloud cover" now included.

Other minor comments are below.

- replace non-tropics with extra-tropics and non-tropical with extra-tropical throughout the manuscript (including in Table 1 and various figure/table captions).

**Changes in manuscript:** Point taken.

- p2, l5: replace 'A radiative forcing broadly refers to' with 'Radiative forcing is'.

**Changes in manuscript:** Point taken.

- p2, l7: ')' is found but it is not preceded by a '('.

**Changes in manuscript:** Change made.

- p2, l15: add reference for estimated chemical lifetime of ozone.

**Changes in manuscript:** Reference of Young et al. (2013, https://doi.org/10.5194/acp-13-2063-2013) is now added.

- p3, l12-13: other authors in the literature have come up with different estimates for global LNOx emissions (see e.g. Martin et al. 2007 and more recently Nault et al. 2017); please add a sentence to recognise other work in this field, which further stresses the large uncertainty on the extent of LNOx emissions.

**Changes in manuscript:** Point taken. We add "Other estimates of global $LNO_x$ emissions include $6 \pm 2$ Tg N yr$^{-1}$ (Martin et al., 2007) and $\sim 9$ Tg N yr$^{-1}$ (Nault et al., 2017)."

- p3, l14: please give more details about the methods used to estimate the direct energy dissipated from lightning.

**Changes in manuscript:** More details are now given in the Supplement S1.

- p6, l2: 'chemistry transport' should be 'chemistry transport models'.

**Changes in manuscript:** Change made.

- p6, l10: 'The upper troposphere is where O3 is most potent as a greenhouse gas' should be rephrased including a description of ozone radiative kernel and adding a reference (see Rap et al. 2015).

**Changes in manuscript:** We have rephrased it as "A tropospheric ozone radiative kernel for all-sky conditions (i.e., clear, cloud overcast, and partially cloudy skies) derived by Rap et al. (2015) suggests that ozone changes in the tropical upper troposphere are up to 10 times more efficient in altering the Earth's radiative flux than other regions."

- p7, l1: replace 'convective component' with 'convection parameterisation scheme'.

**Changes in manuscript:** Change made.

- p9, l25: replace 'this increase' to 'this change'. This is necessary as one of the explanations for 'this increase' in the sentence refers to 'CH4 loss' but methane loss produces a decrease. Therefore it is better to describe this as a 'change'. The next sentence rightly explains the signs of the change in more details.

**Changes in manuscript:** Change made.

- p10, l16: 'The contrast in radiation changes over land the ocean…' should be '…land and ocean…'

**Changes in manuscript:** Change made.

- p17, l19: replace 'the this' with 'this'

**Changes in manuscript:** Change made.

- Fig 4 caption: there are two instances of b). One needs to be replaced with c)

**Changes in manuscript:** Change made.

---

## Author Comment (AC2)

**Reply by the authors to Referee #2's comments on**
**"Radiative impact of improved global parameterisations of oceanic dry deposition of ozone and lightning-generated NO$_x$" (#acp-2022-275)**

**Anonymous Referee #2 (RC2)**

We are grateful to the Referee for taking the time to review our manuscript and making a number of valuable comments, which have significantly improved the quality of the manuscript. In the following, we provide our responses to these comments (the Referee's comments are shown in blue). The locations of the changes made refer to those in the non-tracked version of the revised manuscript.

**Reviewer Summary:** This manuscript uses the ACCESS-UKCA chemistry climate model to investigate the radiative impact of changes to the oceanic dry deposition parameterisation and the lightening NOx (LNOx) parameterisation. The authors find that there is small impact on radiation from the changes to the oceanic dry deposition parameterisation which is attributed to higher tropospheric ozone concentrations. The authors found that the changes to the LNOx parameterization had a relatively large impact on radiation, predominantly in the longwave and over tropical latitudes. These changes were attributed to increased ozone and OH in the upper troposphere, combined with a shorter CH4 lifetime due to the higher oxidant levels.

I have some general and technical comments (please see below) which should be addressed prior to publication.

**General comments:**

1. The manuscript is reasonably clear and well laid out. The manuscript extends the evaluation of a new LNOx parameterization described in Luhar et al. (2021) (as well as the revised oceanic dry deposition parameterization described in Luhar et al., 2018). However, with the evaluation of the new LNOx parameterization split across two publications I found it necessary to read this manuscript in parallel with Luhar et al. (2021) to fully understand how the new parameterization impacts NOx, ozone and OH in the atmosphere, and might therefore be expected to impact radiation. It would be helpful to include a short 'scene setting' paragraph highlighting the main findings from Luhar et al. (2021), with respect to where and by how much the atmospheric composition changed when the new LNOx parameterization was implemented. In my opinion this would be best placed in either the introduction or at the start of Section 3.

**Response:** Point taken. A 'scene setting' paragraph highlighting the main findings from Luhar et al. (2021) is now added in the Introduction (2$^{nd}$ last para) which reads "In most global chemistry models, lightning flash rates used to estimate LNO$_x$ are expressed in terms of convective cloud-top height via Price and Rind's (1992) (PR92) empirical parameterisations for land and ocean. Luhar et al. (2021) tested the PR92 flash-rate parameterisations within ACCESS-UKCA using satellite lightning data and found that while the PR92 parameterisation for land performs well, the oceanic parameterisation underestimates the observed global mean flash frequency by a factor of approximately 30, leading to LNO$_x$ being underestimated proportionally over the ocean. Luhar et al. (2021) improved upon the PR92 flash-rate parameterisations (see section 2.3). They showed that the improved parameterisation for land performs very similar to the corresponding PR92 one in simulating the continental spatial distribution of the global lightning flash rate. The improved

oceanic parameterisation simulates the oceanic and total flash-rate observations much more accurately. Luhar et al. (2021) used the improved flash-rate parameterisations in ACCESS-UKCA and found that they resulted in a considerable impact on the modelled tropospheric composition compared to the default PR92 parameterisations, including an increase in the global $LNO_x$ increased from 4.8 to 6.6 Tg N $yr^{-1}$; an increase in the ozone $O_3$ burden by 8.5%; a 13% increase in the volume-weighted global OH; and a decrease in the methane lifetime by 6.7%. The improved flash-rate parameterisations also led to improved simulation of tropospheric $NO_x$ and ozone in the Southern Hemisphere and over the ocean compared to observations. Luhar et al. (2021) did not examine any changes in aerosol due to the changes in $LNO_x$ (this is done in the present work)."

**Changes in manuscript:** As above.

2. It would be useful to have a paragraph summarizing how the changes in LNOx impact ozone, CH4 and therefore radiation (Net TOA, downward LW, downward SW, surface LW and SW). The changes in radiation shown in Figures 1-4 and Table 1 are described in Section 3.2 and later in that section are attributed to changes in ozone (e.g. in Section 3.2, p. 15, L4-6 '*In terms of the differences from the base run, dry deposition has little effect, but increased LNOx increases the downward flux from ~ 40°N to 40°S presumably due to increased emission of LW by O3 produced by the LNOx.*'). It would be helpful to include a paragraph, possibly in the Conclusions, to summarize the impacts of the changes to the LNOx parameterization on atmospheric composition and how this drives changes in the radiation.

**Response:** A paragraph is added in the Conclusions (2nd paragraph) summarising the impacts of the changes to the $LNO_x$ parameterization on atmospheric composition. It reads "The effects of the $LNO_x$ parameterisation change (which enhanced the $LNO_x$ production from 4.8 to 6.9 Tg N $yr^{-1}$) were a factor of roughly 10 to 20 larger than those due to the dry deposition change. The two combined parameterisation changes increased the global tropospheric $O_3$ burden by 31.9 Tg $O_3$ (11.7%), increased the global $O_3$ column by 3.75 DU (13% of the tropospheric column or 1.2% of the total column), decreased the global mean tropospheric lifetime of $CH_4$ by 0.64 years (8.4%), increased the global column integrated aerosol number concentration by $0.5 \times 10^9$ $cm^{-2}$ (6.7%), and impacted the cloud cover somewhat (zonal mean value by as much as $\pm 0.05$%)." The subsequent paragraphs summarise how radiative fluxes are changed.

**Changes in manuscript:** As above.

3. Are the changes in radiation reported for clear sky or all sky? In Section 3.2, p.15, L7-17 the authors state that downward SW radiative flux at the surface is impacted by clouds through reflection and scattering of solar radiation, but when the new LNOx parameterization is used the increase in downward SW flux from ~20N-60S is attributed to increased ozone. Reporting changes in both the clear-sky and all sky would help to isolate the impact of changes in ozone on the downward SW.

**Response:** Thank you for pointing this out. All radiative fluxes were all sky, and this has been made clear now. We now also report clear-sky fluxes in Table 4 for comparison and add a plot of

the clear-sky downward SW flux in Figure 6. New Tables S1 and S2 in the Supplement report all all-sky and clear-sky radiative fluxes.

Referee 1 also made a comment about the downward SW radiative flux at the surface, noting that the change in this flux cannot possibly be explained in terms of changes in absorption by ozone concentrations in response to changes in $LNO_x$, and that the flux changes could be influenced by cloud and aerosols fields instead. We have now analysed changes in the SW flux at the surface with respect to changes in the modelled aerosol number (or condensation nuclei) concentration and cloud cover. This analysis is reported in a new section 3.4 entitled "Changes in incoming surface shortwave radiation, aerosol and cloud cover" with new plots (Figures 1, 6, 7 and 11). We find that the aerosol number concentration and cloud cover are impacted by changes in $LNO_x$ and that changes in high-level cloud cover can at least partly explain the differences in the shortwave fluxes at the surface.

**Changes in manuscript:** As above.

4. In Section 3.2 the authors find that the spatial distribution of LNOx changes the magnitude of the impact on the radiation. How does the spatial distribution of the radiation change between Runs C and D? Why does the spatial distribution impact radiation differently in Runs C and D?

**Response:** This wasn't quite clear. We have modified the text (in the 2$^{nd}$ para of section 3.3) to read "When the default PR92 lightning flash-rate scheme is used with a uniform global scaling (by a factor of 1.44) so as to give the total global $LNO_x$ the same as that obtained by Run C with the new lightning flash-rate scheme (i.e., 6.9 Tg N yr$^{-1}$), the increase in $R^N_{TOA}$ is 70.9 mW m$^{-2}$. As stated earlier, while the PR92 scheme for land performs very similar to the new scheme in simulating the global spatial distribution of lightning flash rate over land, the oceanic PR92 scheme underestimates the global mean flash-rate distribution considerably over the ocean. Therefore, this uniform scaling of the PR92-derived global flash-rate distribution would cause an over-adjustment of the flash rate (and hence $LNO_x$) over land to compensate for the underestimation by the oceanic parameterisation. Therefore, although the total global $LNO_x$ is the same in both Runs C and D, there is a mismatch in its spatial distribution with Run D having larger $LNO_x$ over land and continue to have lower $LNO_x$ over the ocean than Run C. Thus, the new lightning flash-rate scheme leading to a larger increase in $R^N_{TOA}$ than that obtained by the scaled PR92 scheme implies that how $LNO_x$ is spatially distributed makes a difference in the radiation impact. This difference could possibly be because adding $LNO_x$ to the lower $NO_x$ levels in the marine upper troposphere causes greater ozone production than adding it to the $NO_x$ richer continental upper troposphere, and also because of differences in the photochemical reaction rates as a result of temperature differences over land and sea."

**Changes in manuscript:** As above.

5. It would be helpful for the reader if the units on the figures (generally W m$^{-2}$) were consistent with those used throughout the text, where both W m$^{-2}$ and W m$^{-2}$ are used. Although the exception here would be the right-hand axis in Figures 1 and 2 where the absolute radiative flux is shown for the base run.

**Changes in manuscript:** We agree with the Referee. All units of radiative forcing and flux differences are now in mW m$^{-2}$ (except the IPCC effective radiative forcing units in the 1$^{st}$ paragraph of Introduction which are in W m$^{-2}$) and the units of all absolute radiative fluxes are in W m$^{-2}$.

**Technical comments (by manuscript section):**

Section 2.4

L10: Please revise the sentence to make it a bit easier to follow.

=> ''…the other runs were calculated were indexed as follows on xaxis in relevant plots…'

**Changes in manuscript:** The sentence corrected to read "Differences between the base model run and the other runs were calculated and indexed as follows on the x-axis in relevant plots presented below."

L20-25: Could the authors please add a sentence to explain that Run D is to check the impact of the spatial distribution of the lightning flashes. While this does become clear, it is not immediately apparent as the reader reads through the manuscript.

**Changes in manuscript:** We have modified the sentence.

Section 3

Hopefully Table 1 will appear before Figures 1 and 2 in the typeset manuscript!

**Changes in manuscript:** We have now moved Table 1 (now Table 4) before (now) Figures 2 and 3 in the manuscript. We will also check this in the typeset manuscript.

Section 3.1

L19-26: It would be useful if these abbreviations could be tabulated, ideally within the main body of the text, although an appendix could also be useful.

**Changes in manuscript:** Point taken. The new Table 2 tabulates these abbreviations.

L4-12: I would also suggest that the radiative fluxes for the observed and modelled values (both from ACCESS-UKCA and the CMIP5 ensemble) be tabulated so that the reader can clearly see where the differences and similarities are.

**Changes in manuscript:** Point taken. The new Table 3 tabulates these radiative fluxes.

Section 3.2

L11-13: This statement should be expanded on given that nitrate aerosol forms from both nitrate and ammonium precursors. While the new LNOx parameterisation increases nitrate in the upper

troposphere, ammonium forms from ammonia emission near the surface. The additional nitrate from the new scheme is therefore unlikely to drive any significant increase in nitrate aerosol. However, given the recent availability of a nitrate scheme in UKCA, this should be tested in the future.

**Changes in manuscript:** Point taken. The following text now appears in the last paragraph of section 2.1.

"$LNO_x$ is also a precursor of nitrate aerosol in the upper troposphere, and this aerosol can influence atmospheric radiation (Tost, 2017). However, ACCESS-UKCA as used here does not include nitrate aerosol, which is also the case with most global chemistry-climate models. Of the ten CMIP6 Earth system models that conducted the AerChemMIP (Aerosol and Chemistry Model Intercomparison Project) simulations, only three included nitrate aerosols (Thornhill et al., 2021b). Naik et al. (2021) report that there is a relatively small negative contribution to ERF through formation of nitrate aerosols. Recently, a nitrate scheme has been incorporated in UKCA (Jones et al., 2021) and this should be tested in the future to examine the impact of nitrate aerosol from lightning on radiation. Although the model does not include a nitrate aerosol scheme, the $LNO_x$ changes would impact aerosol through perturbations to background tropospheric oxidants, for example increases in aerosol abundances due to faster oxidation rates of sulfur to sulfate as $LNO_x$ is increased (Murray, 2016)."

L16: Please correct this sentence.

=> 'The contrast in radiation changes over land the ocean is not as stark as that over the tropical and nontropical regions, except for the no-LNOx case.'

**Changes in manuscript:** The sentence has been corrected.

L28: Could the authors please restate what the ozone ERF is? i.e. '…is 18% of the anthropogenic O3 radiative forcing of $0.47 \pm 0.23$ W m$^{-2}$

**Changes in manuscript:** Done.

Figure 4: Pleased include units on the colour bars.

**Changes in manuscript:** Units included.

Section 3.4

L20: '… LNOx parameterisation…' -> '…**the** LNOx parameterisation…'
**Changes in manuscript:** Done.

Conclusions

L3: representation -> represented
**Changes in manuscript:** Done.

L17: '…with ramifications on Earth's radiation budget.' -> '…with ramifications **for the** Earth's radiation budget.'

**Changes in manuscript:** Done.

L11-12: The authors report uncertainty ranges of 241 mW m-2 and 218 mW m-2 – should this be ±241 mW m$^{-2}$ and ±218 mW m$^{-2}$.

**Changes in manuscript:** These values should be ± 121 and ± 109 mW m$^{-2}$, respectively, and have been corrected.